**Challenges and opportunities in modelling savanna ecosystems**
**Authors:** Rhys Whitley[1*], Jason Beringer[2], Lindsay B. Hutley[3], Gabriel Abramowitz[4],
Martin G. De Kauwe[1], Bradley Evans[5], Vanessa Haverd[6], Longhui Li[7], Caitlin Moore[8],
Youngryel Ryu[9], Simon Scheiter[10], Stanislaus J. Schymanski[11], Benjamin Smith[12], Ying-
Ping Wang[13], Mathew Williams[14], Qiang Yu[7]
**Institutions:**
[1]Department of Biological Sciences, Macquarie University, North Ryde, NSW 2109,
Australia
[2]School of Earth and Environment, University of Western Australia, Crawley, WA 6009,
Australia
[3] Research Institute for the Environment and Livelihoods, Charles Darwin University,
Casuarina, NT 0909, Australia
[4]Climate Change Research Centre, University of New South Wales, Kensington, NSW
2033, Australia
[5]Faculty of Agriculture and Environment, University of Sydney, Eveleigh, NSW 2015,
Australia
[6]CSIRO Ocean and Atmosphere, Canberra 2601, Australia
[7]School of Life Sciences, University of Technology Sydney, Ultimo, NSW 2007, Australia
[8]School of Earth, Atmosphere and Environment, Monash University, VIC, 3800, Clayton,
Australia
[9]Department of Landscape Architecture and Rural Systems Engineering, Seoul National
University, Seoul, South Korea
[10]Biodiversität und Kilma Forschungszentrum, Senckenberg Gesellschaft für
Naturforschung, Senckenberganlage 25, 60325 Frankfurt am Main, Germany
[11]ETH Zurich, Department of Environmental System Science, Zurich, Switzerland
[12]Department of Physical Geography and Ecosystem Science, Lund University, Lund,
Sweden
[13]CSIRO Ocean and Atmosphere, Aspendale, Victoria 3195, Australia
[14]School of GeoSciences, University of Edinburgh, Edinburgh, United Kingdom
**\*Corresponding author:**
Name: Rhys Whitley
Email: rhys.whitley@mq.edu.au
Address:  Department of Biological Sciences
Macquarie University
North Ryde, NSW
2109, Australia

**Abstract**

The savanna complex is a highly diverse global biome that occurs within the seasonally dry tropical to sub-tropical equatorial latitudes and are structurally and functionally distinct from grasslands and forests. Savannas are open-canopy environments that encompass a broad demographic continuum, often characterised by a dynamically changing dominance between $C_3$-tree and $C_4$-grass vegetation, where frequent environmental disturbances such as fire modulates the balance between ephemeral and perennial life forms. Climate change is projected to result in significant changes to the savanna floristic structure, with increases to woody biomass expected through $CO_2$ fertilisation in mesic savannas and increased tree mortality expected through increased rainfall interannual variability in xeric savannas. The complex interaction between vegetation and climate that occurs in savannas has traditionally challenged terrestrial biosphere models (TBMs), which aim to simulate the interaction between the atmosphere and the land-surface to predict responses of vegetation to changing in environmental forcing. In this review, we examine whether TBMs are able to adequately represent savanna dynamics and what implications potential deficiencies may have for climate change projection scenarios that rely on these models. We start by highlighting the defining characteristic traits and behaviours of savannas, how these differ across continents, and how this information is (or is not) represented in the structural framework of many TBMs. We highlight three dynamic processes that we believe directly affect the water-use and productivity of the savanna system: phenology; root-water access; and fire dynamics. Following this, we discuss how these processes are represented in many current generation TBMs and whether they are suitable for simulating savanna dynamics. Finally, we give an overview of how eddy-covariance observations in combination with other data sources, can be used in model benchmarking and inter-comparison frameworks to diagnose the performance of TBMs in this environment and formulate roadmaps for future development. Our investigation reveals that many TBMs systematically misrepresent phenology, the effects of fire and root-water access (if they are considered at all) and that these should be critical areas for future development. Furthermore, such processes must not be static (i.e. prescribed behaviour), but be capable of responding to the changing environmental conditions in order to emulate the dynamic behaviour of savannas. Without such developments, however, TBMs will have limited predictive capability in making the critical projections needed to understand how savannas will respond to future global change.

.

## 1    Introduction

Savanna ecosystems are a diverse and complex biome covering approximately 15 to 20% of the global terrestrial surface (Scholes and Hall, 1996) and are important in providing ecosystem services that maintain biodiversity and support the majority of the global livestock (Van Der Werf et al., 2008). Savannas are characterised by a multifaceted strata of vegetation, where an open $C_3$-woody canopy of trees and shrubs overlies a continuous $C_4$-grass understorey and occur in regions that experience a seasonal wet-dry climate, have low topographic relief and infertile soils (Scholes and Archer, 1997). For simplicity, in this paper all woody plants are referred to as trees while grasses include all the herbaceous vegetation. Savanna vegetation structure (defined by the ratio of woody to herbaceous cover) is further modulated by disturbance events (predominantly fire) that create demographic bottlenecks, preventing canopy closure that results in an open, woody system (Scholes and Archer, 1997). Indeed, fire disturbance is a defining characteristic of savannas, particularly for mesic regions (mean annual precipitation (MAP) > 650 mm), potentially holding the ecosystem in a 'meta-stable' state, such that if fire were excluded this open $C_3/C_4$ system would likely shift to a closed $C_3$ forest or woodland (Bond et al., 2005; Sankaran et al., 2005b). The role of fire in modulating vegetation structure allows savannas to occur across a broad demographic continuum, where the density of woody biomass is coupled to the annual amount of rainfall (Hutley et al., 2011; Lehmann et al., 2011). These environmental traits and behaviours therefore mark savannas as one of the most complex terrestrial biomes on the planet, and understanding the vegetation dynamics and underlying processes of this ecosystem type (especially in response to future global change) has proven a challenging task for the ecosystem modelling community(House et al., 2003; Scheiter et al., 2013; Scheiter and Higgins, 2007)

Terrestrial biosphere models (TBMs), aim to predict ecosystem water and carbon transfer between the land-surface and the atmosphere (among other processes) and have mostly underperformed for savanna ecosystems (Whitley et al., 2016). While the reasons for this are in some cases model-specific, a general question can be formed about whether the current generation of TBMs have the predictive capability to adequately simulate savanna dynamics and their response to future global change. Additionally, if limitations do exist, are they a result of an incorrect parameterisation of physical parameters (e.g. root depth, maximum RuBisCO activity, sand and clay soil contents, etc.), the misunderstanding or absence of dynamic biophysical processes (e.g. phenology, root-water uptake, impacts of fire etc.), the challenge of simulating stochastic

events linked to disturbance, or a combination of these? Particular attributes that
characterise savanna environments, such as frequent fire disturbance, the highly
seasonality of available soil-water, and the annual recurrence of $C_4$ grasses (which
except for grasslands are absent in other biomes) are not universally represented in
most model frameworks. While some TBMs have been specifically designed with
savanna dynamics in mind (Coughenour, 1992; Haverd et al., 2016; Scheiter and Higgins,
2009; Simioni et al., 2000), some are simply modified agricultural models (Littleboy and
Mckeon, 1997), with most TBMs attempting to capture savanna dynamics through
calibration to observed time-series data and ad-hoc substitutions of missing processes
(Whitley et al., 2016). Furthermore, little has been done to investigate why simulating
savanna dynamics has fallen outside the scope and capability of many TBMs, such that
these problems can be identified and used in on-going model development.
In this paper we review the current state of modelling for the savanna complex, with
emphasis on how the dynamics and biophysical processes of the savanna ecosystem
may challenge current-generation TBMs. We start with an overview of the global
savanna complex and the many floristic assemblages that fall under this definition. We
discuss how the distinct characteristics, dynamics and regional differences among global
savanna types may have implications for future global change. Then we outline how
some of the defining physical processes of savannas are commonly misrepresented in
TBMs and if these hamper the necessary predictive capability to answer questions on
the future of this biome. Finally, we conclude with a discussion on model evaluation and
benchmarking for this ecosystem, where we argue that eddy flux measurements in
combination with observations from multiple data sources (phenocams, remote-sensing
products, inventory studies) are needed to give a more complete assessment of whether
simulated processes are representative of savanna dynamics.

**2.      The savanna biome**
2.1    *Characteristics and global extent*
At a global scale, biome distributions typically conform to climatic and soil envelopes
and current and future distributions are predictable based on climate and ecosystem
physiology. However, savannas occur in climatic zones that also support grasslands and
forests (Bond 2005, Lehmann et al. 2011), a characteristic that poses major challenges
for TBMs and Dynamic Global Vegetaion Models (DGVMs). Savannas occur across the
tropical to sub-tropical equatorial latitudes occupying a significant portion of the
terrestrial land-surface in seasonal wet-dry climates (Fig. 1). Savannas are therefore
associated with many ecoclimatic regions and are the second largest tropical ecosystem
after rainforests with a global extent of 15.1 million $km^2$, which comprises almost half of
the African continent (Menaut, 1983), 2.1 million $km^2$ of the Cerrado, Campos and
Caatinga ecoregions in South America (Miranda et al., 1997), 1.9 million $km^2$ of the
Australian tropical north (Fox et al., 2001) as well as parts of peninsular India, southeast
Asia (Singh et al., 1985), California and the Iberian peninsula (Ryu et al., 2010a).
While the structure of vegetation in these regions has converged towards a formation of
mixed $C_3$ trees and $C_4$ grasses, the extensive geographical range gives rise to a wide
range of physiognomies and functional attributes with multiple interacting factors such
as seasonality of climate, hydrology, herbivory, fire regime, soil properties and human
influences (Walter, 1973; Walter and Burnett, 1971). Savannas range across tree-grass
ratio from near tree-less grasslands to open forest savanna of high tree cover (Torello-
Raventos et al., 2013). These savanna assemblages can shift to grassland or forest in
response to changes in fire regime, grazing and browsing pressure as well as changing
levels of atmospheric $CO_2$ (Franco et al., 2014) and modelling this structural and
functional diversity is challenging (Moncrieff et al., 2016b). Lehmann et al. (2011)
quantified the different extents of savanna globally, showing that for each continent they
occupy distinctly different climate spaces. For example, South American savannas are
limited to a high but narrower range of MAP (~1000 to 2500 mm), while African and
Australian savannas occur over lower but wider range of MAP (~250 to 2000 mm) and
are further separated by strong differences in interannual rainfall variability and soil
nutrient contents (Bond, 2008). Furthermore, Lehmann et al. (2014) shows that
different interactions between vegetation, rainfall seasonality, fire and soil fertility occur
on each continent and act as determinants of above-ground woody biomass for the
ecosystem.
2.2    *Conceptual models of tree and grass co-existence*
Savannas consist of two co-existing but contrasting life forms; tree and grasses. These
life forms can be considered as mutually exclusive given their differing fire responses,
and shade tolerances as well as their competitive interactions, with grasses
outcompeting trees for water and nutrients when they occupy the same soil horizons
(Bond, 2008) . Ecological theory would suggest exclusion of one or the other and not the
coexistance that is a defining characteristic of savanna (Sankaran et al., 2004). Over the
last five decades numerous mechanisms have been proposed to understand tree-grass
coexistence (Bond, 2008; Lehmann et al., 2011; Lehmann and Parr, 2016; Ratnam et al.,
2011; Scholes and Archer, 1997; Walter and Burnett, 1971). Contrasting conceptual
models have been largely supported by empirical evidence, but no single model has
emerged that provides a generic mechanism explaining coexistence across the three
continents of the tropical savanna biome, Africa, South Amercia and Australia (Lehmann
et al., 2014). Ecological models can be broadly classified into two categories; 1)
competition-based models that feature spatial and temporal separation of resource
usage by trees and grasses that minimises interspecific competition enabling the
persistence of both lifeforms and, 2) demographic-based models where mixtures are
maintained by disturbance that results in bottlenecks in tree recruitment and/or
limitations to tree growth that enables grass persistence.
Root-niche separation models suggest there is a spatial separation of tree and grass root
systems that minimises competition, with grasses exploiting upper soil horizons and
trees developing deeper root systems (i.e. Walter's two-layer hypothesis (Walter and
Burnett, 1971)). Trees rely on excess moisture (and nutrient) draining from surface
horizons to deeper soil layers. Phenological separation models invoke differences in the
timing of growth between trees and grasses. Leaf canopy development and growth in
many savanna trees occurs prior to the onset of the wet season, often before grasses
have germinated or initiated leaf development. As a result, trees can have exclusive
access to resources at the beginning of the growing season, with grasses more
competitive during the growing season proper. Given their deeper root systems, tree
growth persists longer into the dry season, providing an additional period of resource
acquisition at a time when grasses may be senescing. However, grasses are better able
to exploit pulses of resources such as surface soil moisture and nitrogen following short-
term rainfall events, particularly important processes regulating semi-arid savanna
(Chesson et al., 2004). These spatial and temporal separation of resource usage is
thought to minimise competition, enabling co-existence. Other competition models
suggest that tree density becomes self-limiting at a threshold of available moisture
and/or nutrient and are thus unable to completely exclude grasses. These models
assume high rainfall years favour tree growth and recruitment, with poor years
favouring grasses and high interannual variability of rainfall maintaining a relatively
stable equilibrium of trees and grasses over time (Hutley and Setterfield, 2008).
In many savannas, root distribution is spatially separated with mature trees exploiting
deeper soil horizons as the competitive root-niche separation model predicts. In semi-
arid savannas investment in deep root systems may seem counter-intuitive, as rainfall
events tend to be sporadic and small in nature, with little deep drainage. In this case,
surface roots are more effective at exploiting moisture and mineralised nutrients
following these discrete events and shallow rooted grasses tend to have a faster growth
response than trees to these pulse events (Jenerette et al., 2008; Nielsen and Ball, 2015).
Differences in the magnitude and interaction of environmental effects have indicated
savanna vegetation dynamics to be region-specific (Bond, 2008; Bowman and Prior,
2005; Lehmann et al., 2014; Lloyd et al., 2008; Scholes and Hall, 1996), such that there
are marked differences in how regional flora (primarily woody species) have evolved
functional traits to operate within their respective climate space (Cernusak et al., 2011;
Eamus, 1999). For example, major distinctions can be drawn between the savanna flora
of Africa, Australia and South America. Canopies of the African and South American
savanna regions are predominantly characterised by deciduous woody species that are
in most cases (although not always) shallow-rooted and follow a short-term growth
strategy that maximises productivity while environmental conditions are favourable
(Bowman and Prior, 2005; Lehmann et al., 2011; Scholes and Archer, 1997; Stevens et
al., 2017). In contrast, the mesic savanna canopies of northern Australia are dominated
by deep rooted, evergreen *Eucalyptus* and *Corymbia* woody species that favour a long-
term strategy of conservative growth that is insured against an unpredictable climate
(Bowman and Prior, 2005; Eamus et al., 1999, 2001). Consequently, the functional traits
that support deciduous, evergreen or annual strategies have a major impact on the
water and carbon exchange of the ecosystem. For example, Australian mesic savanna
tree canopies operate at almost constant rates of assimilation and transpiration all year
round, due to their deep and extensive root system and ability to make adjustments to
canopy leaf area in times of stress (O'Grady et al., 1999). In these savannas, root
competition between both trees and grass roots in upper soil layers is apparent,
contrary to predictions of niche-separation models that would predict that tree and
grass competition for water and nutrients would be intense. This system serves as an
example of where both root-niche and phenological separation is occurring (Bond,
2008) and highlights the fact that savanna ecosystems cannot be simply reduced to
generalised plant functional type (PFT) and applied globally in land-surface model
(LSM) and dynamic global vegetation model (DGVM) frameworks (Moncrieff et al.,
2016a). One alternative may be to define region-specific PFTs to fully capture the
distinctly different dynamics that are occurring across the ensemble of savanna biomes.
As an alternative conceptual model of tree-grass co-existance, savannas can be viewed
as meta-stable ecosystems, where a range of stable states is possible but the ecosystem
can be deflected from an equilibrium with climate and soil due to a combination of
frequent disturbances (fire and herbivory), resource limitation (soil-moisture and soil
nutrients) and growing conditions, in particular temperature (Lehmann et al., 2014). In
this paradigm, demographic-based models suggest that moisture and nutrient
partitioning is not the sole driver of co-existence and that determinants of tree
demographics and recruitment processes ultimately set tree - grass ratios. Fire,
herbivory and climatic variability are fundamental drivers of tree recruitment and
growth, with high levels of disturbance resulting in demographic bottlenecks that
constrain recruitment and/or growth of woody components and grass persistence
results. At high rainfall sites, in the absence of disturbance, the ecosystem tends towards
forest. High levels of disturbance, particularly fire and herbivory, can push the
ecosystem towards a more open canopy or grassland; this ecosystem trajectory is more
likely at low rainfall sites.
The inherent complexity in savanna function is evident when savanna structure is
correlated with the most significant environmental determinants. Sankaran et al.
(2005a) examined the relationship between tree cover and mean annual rainfall with a
large scatter of tree cover observed at any given rainfall for African savannas. Rainfall
set an upper limit of savanna tree cover, with cover below this due to the interaction of
other determinants such as herbivory, site characteristics (drainage, nutrient
availability, temperature) and fire frequency reducing tree cover and biomass below the
maximum possible for a given rainfall. Lehmann et al. (2011, 2014) took this analysis
further and examined 'savanna-limiting' mechanisms across tropical Africa, Australia
and South America. Their analysis suggested that tropical landscapes consist of mosaics
of closed-canopy forest, savanna and grasslands suggesting that that the limits of
savanna is not simply determined by climate and soils alone. Over the entire range of
environmental conditions in which savannas occur, some fraction of the land surface is
'not-savanna' (Lehmann et al., 2011) suggesting that savannas are not necessarily a
stable state ecosystem and disturbance is required to enable tree and grass co-existence.
It is likely that savanna structure and function results from the interaction of all
processes described above, providing complexity for TBMs, particularly DGVMs that
seek to predict current and future distributions of grassland-savanna-forest transitions
driven by future climatic and/or anthropogenic factors (Scheiter et al., 2013, 2015).
Including both competition and disturbance processes into models can yield more
realistic results for a broad range of rainfall regimes. Competition for moisture between
trees and grasses is a significant factor in maintaining savannas in semi-arid regions
whereas disturbance processes limit tree cover in higher rainfall regions  (Accatino et
al., 2010).
A promising alternative approach of some recent models is to allow savanna
composition to emerge from environmental selection from a mixture of PFTs or trait
combinations, reflecting global diversity in savanna vegetation (e.g. Haverd et al., 2015;
Scheiter and Higgins, 2009; Scheiter et al., 2013; Smith et al., 2001). As an example, the
HAVANA model allows traits such as tree and grass phenology, leaf-area, rooting depth
and relative cover to emerge from incident meteorological variations and their effect on
the evolving ecosystem state (Haverd et al. 2015). Because traits define the response of
the vegetation to climate, it is important that they are themselves adequately
represented in TBMs.

2.2    *The implications of climate change*
Projected global increases in both temperature and the variability of precipitation
patterns as a result of anthropogenic climate change are expected to lead to significant
changes in the structure and diversity of global terrestrial ecosystems (IPCC, 2013;
Wilks Rogers and Beringer, 2017). This will make modelling ecosystem distributions
and biogeochemical fluxes under these transient conditions difficult, challenging TBMs
in how they represent the response of the savanna ecosystem to structural shifts in
vegetation through $CO_2$ fertilisation, increased rainfall seasonality, changes in VPD and
changing fire dynamics (Beringer et al., 2015).
Savannas may be susceptible to small perturbations in climate and could potentially
shift towards alternate closed-forest or open-grassland states as a result (Scheiter and
Higgins, 2009). The total carbon pool of some savannas can be considered as modest
when compared with other ecosystems (e.g. rainforests) (Kilinc and Beringer, 2007).
However, in terms of net primary productivity (NPP), tropical savannas and grasslands
make up a significant proportion, contributing *c.* 30% of annual global NPP (Grace et al.,
2006). A shift in the savanna state towards a more closed system, may lead to these
regions becoming a substantially larger carbon sink (Higgins et al., 2010). Observations
of increased woody vegetation cover (woody encroachment) in many semi-arid
ecosystems and savannas worldwide over recent decades have been attributed to
positive effects of increased atmospheric $CO_2$ on plant water-use effects (Donohue et al.,
2009; Fensholt et al., 2012; Liu et al., 2015). Models suggest that such effects are
predicted to continue in the future. $CO_2$ fertilisation is also expected to favour the more
responsive $C_3$ vegetation, leading to the competitive exclusion of $C_4$ grasses via
supressed grass growth and reduced fire impacts (Bond et al., 2005). Model projections
by Scheiter and Higgins (Scheiter and Higgins, 2009), and Higgins and Scheiter (Higgins
and Scheiter, 2012) suggest future range shifts of African savanna into more arid
climates as a consequence of elevated $CO_2$, with concurrent transformation of current
savanna habitats to forests under a stationary rainfall assumption. Recent evidence
underscores the significant role of savannas in the global carbon cycle (Ahlström et al.,
2015; Haverd et al., 2016; Poulter et al., 2014).
The response of savanna structure and function to changes in precipitation patterns is
highly uncertain (Wilks Rogers and Beringer, 2017). Scheiter et al. (2015) investigated
the effect of variable rainfall seasonality, projecting modest to large increases in above-
ground biomass for savannas in northern Australia. The authors showed that woody
biomass in this region increased despite significant changes to precipitation regimes,
being predominantly driven by $CO_2$ fertilisation and rainfall seasonality determining the
magnitude of the increase (Fig. 2) (Scheiter et al., 2015). However, some studies have
indicated that while increased rainfall seasonality may have a small effect in mesic
savanna systems, it may potentially act as an opposing effect to woody encroachment in
semi-arid savanna systems (Fensham et al., 2009; Hiernaux et al., 2009). For example,
Fensham et al. (Fensham et al., 2009) have shown significant tree mortality to occur as a
result of drought in a semi-arid savannas in south-west Queensland, suggesting that
severe water-stress may counteract the positive effect of $CO_2$ fertilisation on ecosystem
carbon balance. Alternatively, forest dieback as a result of increased rainfall seasonality
and more frequent drought occurrence may lead to an expansion of savanna distribution
in some regions. For example, simulations of the Amazon basin have projected a
possible conversion of rainforest to savanna in eastern Amazonia as a result of forest
dieback induced by severe water stress and fire disturbance (Cox et al., 2004; Malhi et
al., 2009).
Increased warming and changes to rainfall seasonality are expected to alter the
interaction between climate, fire and savannas in the future (Beringer et al., 2015),
however, we leave discussion of savanna fire dynamics and the ability of TBMs to
simulate this process until later in this paper. Permanent shifts in the structure and
physiology of the savanna complex as a result of climate change is expected to have a
major impact on the exchange of water, energy and carbon that occurs in this system,
which in turn ultimately affects global biogeochemical cycling and climate (Beringer et
al., 2015; Pitman, 2003).

**3. The capability of models to simulate savanna ecosystems**
The term '*terrestrial biosphere model'* refers to a variety of bottom-up modelling
approaches that simulate coupled dynamics of water, energy, carbon, and in some cases
nutrients in vegetation and soils. TBMs range from stand models, which simulate
specific ecosystems in high detail, up to DGVMs, which can simulate ecosystem
composition, biogeochemical processes and energy exchange and the spatial
distribution of multiple ecosystems at regional to global scales. Consequently, TBMs
collectively operate over different temporal and spatial scales and employ processes of
different scope in simulating ecosystem dynamics. However, common to all TBMs are
that they are governed by the same biophysical principles of energy and mass transfer
that determines the dynamics of plant life (Pitman, 2003). Consequently, the predictive
capability of different TBMs at determining the exchange of water, energy and carbon
between the surface and atmosphere should be convergent within a reasonable degree
of error (Abramowitz, 2012). However, model intercomparison and benchmarking
studies have shown that many TBMs are unable to meet reasonable levels of expected
performance as a result of a systematic misrepresentation of key ecosystem processes
(Abramowitz et al., 2008; Best et al., 2015; Blyth et al., 2011; Mahecha et al., 2010).
The misrepresentations of ecosystem processes is particularly evident in savannas, for
which many TBMs have not been developed for, nor tested on (Baudena et al., 2015;
Cramer et al., 2001; Whitley et al., 2016) . Seasonal competition and access to
belowground resources (soil moisture and nutrients), impacts of browsing and grazing,
and stochastic disturbance events (fire), are less prevalent in other ecosystems and are
therefore not well represented (or even missing) in many TBMs (House et al., 2003;
Whitley et al., 2016). Other stochastic events common in savanna environments are
precipitation pulses that in semi-arid savanna, drive production and respiration
processes (Huxman et al., 2004; Williams et al., 2009). High spatial and temporal
variability of pulse events, coupled with the differential responses of tree and grasses
complicates application of TBMs in savannas. Precipitation pulses are particularly
significant in semi-arid ecosystems and pulse size determines the relative response of
ecosystem respiration (Re) and gross primary production (GPP), with large events
driving high rates of Re that proceede any response in GPP and the ecosystem may
switch to source of $CO_2$ to the atmosphere for a period post event (Huxman et al., 2004).
The annual C balance can be determined by the frequency, magnitude and duration of
pulse events (Cleverly et al., 2013).
Conventional TBMs still lack this capability and tend to underestimate Re and
overestimate Ra in semi-arid regions (Mitchell et al., 2011) and therefore have limited
application for biomes in the seasonally dry tropics, which in turn becomes a large
source of uncertainty in future global studies (Scheiter and Higgins, 2009). However, we
believe that incorporating key processes that drive savanna dynamics into current-
generation TBMs has great potential, considering that even small modifications can lead
to large gains in performance (Feddes et al., 2001; Whitley et al., 2011). It is clear from
the above background and discussion that the ecological processes in savannas are
numerous, detailed, complex and important as they can all have differential responses to
environmental drivers.  We suggest that most of the detailed ecological processes
become emergent properties within model frameworks.  Therefore we do not attempt to
capture everything but rather we have identified phenology, root-water uptake and fire
disturbance as three critical processes in savannas that deserve special consideration in
modern TBMs as explained below.
*3.1 Phenology*
Phenology is an expression of the seasonal dynamics of the structural vegetation
properties that define their growing season and ultimately their productivity (Moore et
al., 2016a). Here we limit our discussion to the phenology of leaf cover. In seasonally dry
climates phenology is driven by soil-moisture availability, and the length of the growing
season for shallow rooting plants is determined by the seasonality of rainfall (Kanniah et
al., 2010; Ma et al., 2013; Scholes and Archer, 1997). Plants respond differently to water
availability, such that phenology is a function of the dominant species within the
ecosystem. Deciduous trees and annual grasses are photosynthetically active during the
wet season only and respectively senesce or become dormant at the beginning of the dry
season, while evergreen trees may remain permanently active throughout the year,
potentially responding to soil-moisture depletion by gradually reducing their canopy
leaf area (Bowman and Prior, 2005). These dynamics are critically important, as they
control the amount and seasonality of carbon-uptake and water-use. In TBMs,
ecosystem phenology is typically represented in one of two ways. The first is via direct
*prescription* of this information as an additional input to the model, where observations
of leaf area index (LAI) (either in-situ measurements or satellite derived products) are
used to express the change in ecosystem canopy cover over time (Whitley et al., 2011).
The second is as a *prognostic* determination using a growth sub-module, where carbon
allocation and leaf metabolic activity are simulated and dependent upon the time-
varying conditions of temperature and soil-water availability (Scheiter and Higgins,
2009). Prescription of phenology from observed LAI dynamics requires an accurate
determination of the separate tree and grass components from bulk ecosystem LAI to be
feasible for savanna ecosystems (Whitley et al., 2011). In many cases, this separation is
assumed to be static, ignoring the different seasonal changes in tree and grass cover
over time (Scholes and Archer, 1997). In fact, no models that we are aware of
dynamically partition LAI as it is prescribed. Donohue et al. (Donohue et al., 2009) offers
an *a priori* method that can determine separate tree and grass LAI signals. This method
assumes that the high variability in the bulk signal is attributed to herbaceous
vegetation, such that the remaining, less variable signal is attributed to woody
vegetation (Fig. 3). A prescription of separate tree and grass LAI inputs was found to be
necessary for simulating water and carbon exchange for a mesic savanna site in
northern Australia (Whitley et al., 2011), and in determining a reduced error estimate of
the Australian continental water and carbon balance (Haverd et al., 2013) to which
savannas contribute significantly. The major drawback to prescribing LAI as a model
input is that the model's scope is limited to hindcast applications. Because this
information is supplied to the model, the floristic structure and its evolution over time is
fixed, and cannot respond to changing environmental conditions (e.g. shifts in
precipitation patterns) that are likely to have an impact on the tree-grass demography
(Ma et al., 2013). Consequently, a dynamic approach where savanna phenology is
explicitly simulated and dynamically responds to climate and disturbance offers a more
promising path forward.
Allocation-growth schemes allow models to express phenology in terms of the evolution
of carbon investment in leaf area over time, limited by the availability of resources for
growth (Haverd et al., 2016). These schemes effectively work by distributing assimilated
carbon (via net primary productivity; NPP) to the root, stem and leaf compartments of
the simulated plant, where allocation to the leaf is dependent on the plant being
metabolically active or dormant (Cramer et al., 2001). In some TBMs, allocation to these
compartments is a fixed ratio (set according to plant functional type) and metabolic leaf
activity is defined through a set of threshold bioclimatic indicators (e.g. photoperiod,

moisture availability and temperature) that determine whether conditions are favourable for photosynthesis (Jolly et al., 2005). However, more recent advances use an alternative approach of dynamically guiding allocation towards the compartment that most limits a plant's growth (Scheiter and Higgins, 2009) or dynamically optimising daily allocation, to maximise long-term NPP and control the competitive balance between trees and grasses (Haverd et al., 2016). The latter approach, based on optimality theory (Raupach, 2005), is related to the approach followed by Schymanski et al. (2009), who assumed that vegetation dynamically optimises its properties (root system and foliage) to maximise its long-term net carbon profit. These approaches, which assume a more dynamic coupling between allocation and phenology, allow plant form and community structure to evolve in response to changes in resource availability (light, water or carbon) over time, with phenology becoming an emergent property of this process. Dynamic allocation schemes enable a TBM to answer questions regarding how changing climate or elevated atmospheric $CO_2$ concentrations may alter structural properties of the ecosystem, and the resultant feedbacks on water, carbon and energy cycling (Scheiter and Higgins, 2009; Schymanski et al., 2015).

*3.2 Root-water access and uptake*

The root zone is critically important in maintaining water and carbon fluxes, as it defines an ecosystem's accessible belowground resources and vulnerability to prolonged dry periods (De Kauwe et al., 2015). Savannas occur in  seasonally dry climates where productivity is primarily limited by dry season water-availability (Kanniah et al., 2010, 2011, 2012), which is largely determined by plant regulation of water transport (through leaf stomatal conductance and stem capacitance) and the root zone water storage capacity and access (distribution of fine root biomass (Eamus et al., 2002). Co-ordination of the whole soil-root-leaf-atmosphere pathway in response to the highly seasonal climate is critical to the survival of savanna plants and is intrinsically linked to their phenology. Partitioning of root water uptake is a key component of competition models describing tree-grass co-existence as described above. For example, deciduous and annual savanna species have shallow root profiles (approx. 0.5 to 2 m) and highly conductive vascular systems to maximise productivity during the wet season (February and Higgins, 2010). In contrast, evergreen savanna species invest in highly regulated hydraulic architectures and deep root systems (> 2 m) that can access deep soil water stores to maintain continuous productivity throughout the dry season

(Bowman and Prior, 2005). It is therefore critically important that the specific root
system and hydraulic architectures of savanna species be adequately represented in
models to simulate water and carbon fluxes of this system.
Soil and plant hydraulic traits such as rooting depth and distribution, stem hydraulic
resistance, and sand and clay contents are typically represented as fixed parameters in
TBMs. Of these traits, the root profile acts as the first-order control on soil-water supply
and therefore determines the capability of a simulated plant to remain active through
rain-free periods (Eamus et al., 1999). The root profile within a soil column is generally
modelled as an exponentially declining root-surface area with depth, the limit of which
extends to some prescribed level. Although some models are capable of dynamically
determining the size of the root profile as an emergent property of productivity and
climate (e.g. Haverd et al., 2016; Schymanski et al., 2009), more typically, the maximum
rooting depth is fixed at approximately 1.5 to 2.0 m (Whitley et al., 2016). However,
studies have shown that woody plants in semi-arid or seasonally dry climates
(particularly those in Australia) exhibit deep root systems to remain active during
prolonged dry periods (Duursma et al., 2011; Hutley et al., 2000; O'Grady et al., 1999).
Numerous modelling studies have shown that a rooting profile of significant depth (> 2
m) is required to achieve good model-data agreement (Fisher et al., 2007; Haxeltine and
Prentice, 1996; Schymanski et al., 2009; Whitley et al., 2016, 2011). While
characterisation of the rooting depth in savanna modelling exercises may be seen as a
matter of correct parameterisation rather than one of systematic process, its role as a
first-order control on water supply in seasonally water-limited systems gives it
significant weight in the overall determination of carbon uptake. Furthermore, long-
term responses of rooting depth to climate change or elevated atmospheric $CO_2$
concentrations may substantially alter structure, resource use and carbon uptake of
savanna ecosystems (Schymanski et al., 2015). Consequently, rooting depths that
sufficiently represent either deciduous or evergreen tree species need to be considered
when modelling savannas.
Directly coupled to the characterisation of the root-zone is the systematic process by
which soil-water is extracted by the root system. The process of root-water uptake in
TBMs has been simulated using numerous schemes. One approach assumes that the
amount of extracted water by roots is a function of the root density distribution within
the soil column and is expressed through an additional sink term to the Richard's
equation, which represents the flow of water in an unsaturated soil (Wang et al., 2011).
In such schemes, root-water uptake may be weighted by the distribution of fine-root
biomass in the soil, such that soil-layers with the greatest density of fine-root biomass
largely determine the soil-water status of the plant, its stomatal behaviour, and
therefore its sensitivity to soil drying (Wang et al., 2011). The exponential decay
function conventionally used to describe the root profile in most TBMs (an exception is
Schymanski et al. 2009) can result in simulated stomatal behaviour that is heavily
weighted towards the moisture content of the upper soil profile, making them highly
sensitive to drought (De Kauwe et al., 2015). In reality, the active root distribution of
savannas is not static, nor so limited, but responds dynamically to wherever water is
available. For example, eucalypts occurring in Australian mesic savannas invest in 'dual-
root' systems that are capable of switching their root activity between subsurface and
subsoil respectively to access water continually during both wet and dry seasons (Chen
et al., 2004). Alternative root-water uptake schemes do exist that describe a more
dynamic response to long-term changes in soil conditions. One such scheme by Williams
et al. (2001) considers root activity to change over time and be concentrated towards
parts of the root zone where the plant can sustainably extract the maximal amount of
available water. Consequently, this scheme effectively weights soil-water status over the
distribution of fine-root biomass, such that simulated root-water uptake dynamically
responds to the wetting and drying of the soil profile over time (Fig. 4). Another
alternative approach by Schymanski et al. (Schymanski et al., 2008) allows the root zone
to dynamically adjust the vertical distribution of root biomass in the profile to balance
canopy water demand while minimising structural costs of maintaining such a root
system . These alternate schemes offer a more dynamic approach to modelling the
hydraulic architecture of species occurring in savannas and other semi-arid ecosystems,
and have demonstrated high predictive skill in these environments (Schymanski et al.,
2008, 2009; Whitley et al., 2011). Therefore, given the distinct seasonality of savanna
ecosystems, dynamic root-water extraction schemes are needed to simulate how the
root zone responds to the evolution of soil-water supply over time.
In should be noted that the above discussion on root-water uptake is one based on
relatively simple model processes, however, savanna ecosystems have much more
complex interactions across the soil-root-stem-leaf-atmosphere continuum. Additional
processes such as adaptive changes in root architecture across seasonal and interannual
timescales, rhizosphere-root interactions, hydraulic redistribution, plant stem water
storage and limitations on leaf function due to water demand across soil-root-stem-leaf-
atmosphere continuum (Lai and Katul, 2000; Steudle, 2000; Vrugt et al., 2001) may also
be important in simulating root water uptake.
*3.3 Disturbance*
Ecosystem structure and function in seasonally dry tropical systems such as savanna, is
strongly shaped by environmental disturbance, such as persistent herbivory pressures,
frequent low-impact fire events, and infrequent high-impact cyclones (Bond, 2008;
Hutley and Beringer, 2011) that shape tree demographics. Fires have a significant
impact on land-surface exchange and vegetation structure and contribute to greenhouse
gas emissions through the consumption of biomass (Beringer et al., 1995, 2015). Fire
has the capacity to alter land-surface exchange fluxes through the removal of functional
leaf area (reduced LAI) and the blackening of the surface (reduced albedo), temporarily
reducing net carbon uptake (Beringer et al., 2003, 2007) and altering the atmospheric
boundary layer to affect convective cloud formation and precipitation (Görgen et al.,
2006; Lynch et al., 2007). Regarding vegetation structure, fire influences the competitive
balance between tree and grass demographics, suppressing recruitment of woody
saplings to adults, thereby deflecting the system from reaching canopy closure (Beringer
et al., 2015; Higgins et al., 2000). Work by Bond et al. (Bond et al., 2005) underlines the
potential effect of removing fire from the savanna system, with substantial increases in
woody biomass and major structural shifts towards closed forests. This is further
supported by more empirical studies involving fire exclusion experiments and showing
similar tendencies towards woody dominance (Bond and Van Wilgen, 1996; Scott et al.,
2012). Given that future climate projections point to predict higher temperatures and
less precipitation for sub-tropical regions (Wilks Rogers and Beringer, 2017) the
representation of short- and long-term impacts of fire on savanna structure and function
in TBMs may be important in understanding how savanna landscapes may respond to
changes in fire frequency and intensity (Bond et al., 2005).
Fire is commonly simulated as a stochastic process, with the probability of occurrence
increasing with the accumulation of litterfall and grass biomass (fuel loads), combined
with dry and windy environmental conditions that promote ignition (generally through
lightning) (Kelley et al., 2014). The simulated amount of biomass consumed after an
ignition event differs among models. Recent advances in simulating savanna fire
processes have led to more complete representations of the complex interaction
between fire and woody vegetation and how this shapes savanna structure. For
example, Scheiter and Higgins (2009) consider a 'topkill' probability that supresses
woody plant succession if fire intensity is of a critical magnitude determined by the
plant's fire-resisting functional traits (e.g. height, stem diameter, bark thickness). This
scheme allows fire to directly shape the savanna tree population through the dynamics
of woody establishment, resprouting and mortality. Additionally, Kelley et al. (2014)
have similarly considered how fire-resisting functional traits of woody vegetation alter
the fire dynamics of seasonally dry environments. It should be noted that both studies
do not consider anthropogenic ignition events, whereas recent work by Scheiter et al.
(Scheiter et al., 2015) suggests that fire management can be simulated using fixed fire
return intervals.
Many TBMs simulate fire as an instantaneous event through emissions and removal of
biomass, but may not consider the transient effects that fire has on land-surface after the
event has occurred. It has been demonstrated previously that these post-fire effects on
canopy surface mass and energy exchange can be significant, with fire indirectly
supressing productivity by *c.* 16% (+0.7 tC ha$^{-1}$ yr$^{-1}$) (Fig. 5) (Beringer et al., 2007).
During this period, resprouting rather than climate drives productivity, with respiration
exceeding photosynthesis as a result of the regenerative cost of replacing damaged or
lost stems and leaf area (Cernusak et al., 2006). In fact many modelling analyses of
savannas dynamics have removed the post-fire periods completely from any assessment
of performance, such that evaluation has been limited to periods where the model is
considered to be 'fit for purpose' (Whitley et al., 2016, 2011). Fire is an integral part of
savanna dynamics; it is important to include fire events in the analysis of savanna
carbon and water fluxes or model performance. Furthermore, an accurate and robust
representation of fire effects on savanna ecosystems is needed to answer questions
about how savanna dynamics may change under future climate scenarios, as fire
regimes have significant impacts on the carbon balance of these systems (Beringer et al.,

613   2015).

Other disturbance processes such herbivory pressures and impact of cyclones have
limited to no representation in models. The removal of aboveground biomass through
grazing and browsing, is commonly represented as a set fraction of cover or productivity
that is removed over time according to the degree of local agricultural pressures, but has
been represented dynamically in some models (e.g. Pachzelt et al., 2015). Grazing and
browsing are of central importance in many of the world's savannas and like fire,
strongly influence cover and productivity (Bond and Keeley, 2005). The importance of
herbivory as a determinant varies between savanna regions, and appears to largely
reflect the abundance of large herbivores present. In parts of Africa, woody vegetation
density has sometimes been reduced by large herbivores, for example uprooting of trees
by elephants when browsing (Asner et al., 2016; Laws, 1970).
Bond and Keeley  (2005) suggested that browsing is analogous to fire as once saplings
escape a flame or browsing height, they are beyond the reach of most mammal
herbivores. Invertebrates are also significant herbivores, particularly grasshoppers,
caterpillars, ants and termites. Mammal herbivores are typically categorized as grazers,
browsers or mixed feeders, who can vary their diet depending on food availability.
Large herbivores can lead to changes in species composition, woody vegetation density
and soil structure.  Browsers such as giraffes can reduce woody seedling and sapling
growth thereby keeping them within a fire-sensitive heights for decades. Reductions in
grass biomass following grazing leads to a reduction of fuel and thus fire frequency and
intensity, enhancing the survival of saplings and adult tress (Bond, 2008). Fire also
affects herbivory as herbivores may favour post-fire vegetation regrowth.
Termite pressures have also been shown to supress productivity (Hutley and Beringer,
2011), but this loss may be too small to be considered as a significant consumer of
biomass in TBMs. No models that the authors are aware of simulate the effect of
cyclones on vegetation dynamics in tropical systems despite their impact on long-term
ecosystem structure and productivity. Cyclones are infrequent but high impact
disturbance events that occur in any mesic savanna that lies close to the coastline, and
can effectively 'restart' the savanna system through the mass removal of woody biomass
(Hutley et al., 2013). Hutley and Beringer (2011) have shown that for an Australian
mesic savanna, a bimodal distribution of the tree class sizes at the site indicates two
major recruitment events that corresponds with two of the last great cyclones to occur
in the region. Despite the immediate and significant loss of woody biomass during those
events, recovery was possible and pushed this site to a carbon sink over many decades.
Despite the impact that cyclones have on savanna structure it is somewhat understated
in the literature, possibly due to the integrated loss in productivity over long-periods
being small (Hutley et al., 2013) as well as the difficulty in simulating cyclone frequency
and intensity across the landscape at present or in the future. However, we believe
because cyclones modulate savanna structure so strongly, there is a need them to be
considered in TBM frameworks, particularly for long-term projections on productivity.
While few models have the capability to simulate the full spectrum of environmental
disturbance effects on savanna ecosystems explicitly, the significant modulating impact
they have on savanna structure and function flags these processes as a high priority in
future model development.

**4. Testing and developing models for application in savannas**

Given that there are strong indications that critical savanna processes are likely misrepresented in current-generation TBMs, there is a clear need for further model testing and evaluation to be conducted for this ecosystem. Savannas have been the subject of improved research over the past two decades, resulting in a good and evolving understanding of their complicated structure, function, and contribution to global biogeochemical cycling (Higgins and Scheiter, 2012; Lehmann et al., 2014; Sankaran et al., 2005b; Scholes and Archer, 1997). Despite this, our increased understanding of savanna dynamics has not been properly translated into many modern TBMs, with the effect of major deficiencies in modelling this ecosystem (Whitley et al., 2016). Consequently, there is still a great necessity for continuous, consistent and objective studies to test and develop how savanna dynamics are represented and simulated. Below we highlight how datasets from multiple sources that include eddy flux towers, satellites, and *in situ* studies can inform model development and be used in evaluation and benchmarking studies.

*4.1 Datasets to inform model development*

Eddy-covariance (EC) systems that observe the instantaneous response of water, energy and carbon exchange to variability in climate and the evolution of this response over time provide crucial information on which to test and develop TBM application in savanna ecosystems (Beringer et al., 2016a, 2016b). Turbulent fluxes measured by EC systems that include net ecosystem exchange and latent and sensible heat are common model outputs, such that this information is commonly used to validate TBMs. Local meteorological forcing (e.g. short-wave irradiance; SW, air temperature, rainfall, etc.) that is concurrently measured with the turbulent fluxes by other instruments (rainfall and temperature gauges, radiation sensors, etc.) are common model inputs and are used to drive TBMs. Additionally, both turbulent fluxes and meteorological forcing are measured at the same temporal and ecosystem scale at which TBMs are commonly run (Aubinet et al., 2012). Consequently, these datasets offer an unparalleled capability in diagnostic model evaluation (Abramowitz, 2012; Balzarolo et al., 2014; Mahecha et al., 2010). The use of EC datasets to evaluate TBMs and inform further development has been a long running practice within the ecosystem modelling community, with particular success being reported for some savanna studies in Australia (Barrett et al.,

2005; Haverd et al., 2013, 2016, Schymanski et al., 2007, 2009, Whitley et al., 2016,
2011). Here we outline two opportunities of using EC systems in assessing model skill
for savanna ecosystems.
The first of these addresses the problem that EC datasets represent the integrated sum
of turbulent fluxes for the entire system (i.e. soil, grass, shrubs and trees) that are not
easily separated. Assessing model performance using bulk measurements does not
consider the separate responses of the functionally different $C_3$ tree and $C_4$ grass
components that respond differently to climate (Whitley et al., 2016, 2011). However, a
recent study by Moore et al. (Moore et al., 2016b) has shown for a mesic savanna site in
Australia that separate observations of canopy and understorey fluxes can be
determined by using a 'dual tower' EC system that observes turbulent fluxes at
reference points above and beneath the canopy (Fig 6). Datasets such as this provide a
valuable resource to analyse the skill of separate model processes, i.e. simulation of tree
and grass leaf gas-exchange, and test the degree of model equifinality (Bevan and Freer,
2001) at predicting the bulk ecosystem flux. A further collection of coupled over- and
understorey EC datasets is therefore critically needed to verify that simulated tree and
grass dynamics are correctly represented in TBMs.
The second opportunity addresses the issue of savanna landscape heterogeneity.
Savannas are not a homogeneous PFT, but rather a continuum of changing tree and
grass demographics that shift biogeographically with rainfall and other factors (Ma et
al., 2013). Ecological gradient studies, such as the Kalahari Transect (Scholes et al.,
2004) and North Australian Tropical Transect (NATT) (Hutley et al., 2011), have shown
turbulent fluxes along a declining rainfall gradient to be strongly linked to structural
changes in vegetation (Beringer et al., 2011a, 2011b). In essence, the spatial response to
a systematic changes in rainfall (or other resources or disturbance intensities)
represents the possible future temporal response to changing climate, such that
transects can be used to evaluate TBMs by their ability to emulate the full spectrum of
savanna behaviour rather than at just one point. A recent model intercomparison study
by Whitley et al. (2015) used turbulent flux observations sampled along the NATT to
evaluate a set of six TBMs, and documented only poor to moderate performance for
those savanna sites. Model evaluations studies that test model predictive skill across
both time and space are therefore crucial to projecting how savannas dynamically
respond to changing climate.

While EC systems provide valuable datasets on which to test and develop models, they are unable to provide a complete evaluation, as they cannot completely capture long-term temporal and spatial scale features (e.g. demographic structural shifts in vegetation), nor provide detail on underlying ecosystem processes (e.g. root-water dynamics and carbon allocation) (Abramowitz, 2012; Haverd et al., 2013; Keenan et al., 2012). Additional sources of data and their collection are therefore critical to informing how well models are representing the specific dynamics that unique to savannas. Model inversion studies have shown EC datasets give significant constraint to predictions of NPP, however extra ancillary data that is informative of other underlying processes was required to further constrain uncertainty (Haverd et al., 2013; Keenan et al., 2012). Here, we suggest how each of the three critical savanna processes highlighted in this paper can potentially be tested in addition to EC datasets. Satellite derived estimates of remotely sensed near-surface reflectance (Ma et al., 2013; Ryu et al., 2010b) and digital imagery from 'PhenoCams' (Moore et al., 2016a; Sonnentag et al., 2012), provide a good resource for testing simulated phenology, particularly the 'green-up' and 'brown-down' phases. Additionally, Advanced Very High Resolution Radiometer (AVHRR) data can provide 'burnt area' maps that quantify the frequency of fire events, which can inform the probability of occurrence in simulated fire-dynamics. Above- and belowground carbon inventory studies (Chen et al., 2003; Kgope et al., 2010) provide highly valuable sources of information in how plants allocate their resources for growth, which can test the efficacy of TBM allocation scheme. Digital soil maps also provide an excellent resource in parameterising simulated soil profiles (e.g. Isbell, 2002; Sanchez et al., 2009). However the spatial resolution of these data products can be coarser than operating resolution of many TBMs, such that site-level measurements should be used when possible. Excavation studies that quantify savanna tree root-systems (Chen et al., 2004) and soil-moisture probes installed to greater depths (> 2 m) are informative about the evolution of the soil-root zone over time (e.g. surface root density, root depth), and such data may be critical to understanding whether current root-water extraction schemes in TBMs are capable of simulating the dry season response of savanna tree species (Whitley et al., 2016). Other useful approaches for elucidating how and where plants gain their water, include sap flow measurements (Zeppel et al., 2008), gas chambers (Hamel et al., 2015) and soil-plant-water experiments (Midwood et al., 1998). In additional, hydrogen and oxygen stable isotope ratios of water within plants provide new information on water sources, interactions between plant species and water use patterns under various conditions (see review by Yang et al. (2010)).

Finally, localised observations of plant traits such leaf-mass per area, stomatal
conductance ($g_s$), tree height, etc. are needed to inform a better parameterisation of
savanna specific PFTs (Cernusak et al., 2011). For example, specific leaf-level
information such as Rubisco activity ($V_{cmax}$) and RuPB regeneration ($J_{max}$) for both $C_3$ and
$C_4$ plants are critically needed to inform the Farquhar leaf photosynthesis models
(Farquhar et al., 1980), while information on $g_s$ and leaf water potential ($\Psi_{leaf}$) are
important in parameterising the many stomatal conductance models used in TBMs (Ball
et al., 1987; Medlyn et al., 2011; Williams et al., 1996). Leaf capacitance and water
potential data are also critically important in characterising model sensitivity to drought
(Williams et al., 2001), but this information is severely lacking for savannas.
Given that there are many interacting effects occurring in savannas, an integration of
multiple data sources is therefore necessary for a more complete evaluation of how well
TBMs perform in this environment. We recommend that future EC studies, particularly
along transects as mentioned above, should include intensive field campaigns that are
targeted towards a more complete characterisation of the site. This would include root
excavations and the collection of plant trait measurements that sample such data within
the footprint of an EC tower. Collaborative research networks, such as those of TERN
(Terrestrial Ecosystem Research Network), NEON (National Ecological Observatory
Network) and SAEON (South African Environmental Observation Network) that have
the resources and infrastructure to conduct such campaigns will be needed to meet
these demands for more observational data.

*4.2 Model evaluation and benchmarking*
Multiple dynamic processes drive savanna structure and function, and an understanding
of the causes and reasons for why TBMs systematically misrepresent this ecosystem is
paramount to future development. Consequently, a complete diagnostic evaluation of
model performance in savanna ecosystems requires more than just simple model-model
and model-data comparisons where 'good performance' is determined from a score in a
given metric (e.g. a high correlation between observed and predicted values). Instead
evaluation should also consider parsimony, physical representativeness and 'out-of-
sample' capability of the model itself (Abramowitz et al., 2008). A holistic evaluation of
the biophysical, biogeochemical and ecological processes represented in TBMs has
therefore been the aim of many international model intercomparison projects, with

some notable examples being the Project for the Intercomparison of Land surface Parameterization Schemes (PILPS) (Pitman, 2003) and the Coupled Carbon Cycle Climate Model Intercomparison Project (C4MIP) (Friedlingstein et al., 2006). Most recently the International Land Model Benchmarking Project (ILAMB) has been established to holistically assess the major components of TMBs, through a model-data comparison framework that utilises standardised benchmarking and performance metrics to identify critical model deficiencies and guide future development (Luo et al., 2012). A major goal of ILAMB is to support the development of open-source software that can facilitate such a benchmarking framework by the international modelling community. The Protocol for the Analysis of Land-Surface models (PALS; http://www.pals.unsw.edu.au/) has been recently developed to meet the formalism outlined by ILAMB, using standardised experiments to benchmark TBMs in terms of how well they should be expected to perform, based on their complexity and the information used to drive them (Abramowitz, 2012). In brief, PALS uses a set of empirical benchmarks to fulfil the role of an arbitrary TBM of increasing complexity by quantifying the amount of information in the meteorological forcing useful to reproduce water, carbon and energy exchange. This gives a point of reference to measure at what level of complexity a TBM is performing, by comparison of the statistical performance between model and benchmark (Best et al., 2015). For example, we can assess whether a sophisticated, state-of-the-art DGVM can outperform a simple linear regression against shortwave irradiance (SW) at predicting GPP. If the outcome of this test were negative, then this may suggest that the model does not capture the sensitivity of GPP to SW accurately, flagging it as a priority for investigation and development. The important distinction to make with the benchmarks is that they have no internal state variables such as soil moisture and temperature, nor any knowledge of vegetation or soil properties; they represent a purely instantaneous response to the meteorological forcing (Abramowitz et al., 2008). The protocol of PALS meets the four criteria outlined by ILAMB that objectively, effectively and reliably measure the underlying processes of a TBM to improve its predictive skill (Luo et al., 2012). A direct application of this protocol was presented in a model intercomparison study by Whitley et al. (Whitley et al., 2015), where they assessed the predictive capability of TBMs in savanna ecosystems by comparing model outputs to 3 simple empirical benchmarks. In this study the authors used 6 calibrated TBMs to predict ecosystem latent energy and GPP at five savanna sites along the NATT, and found that in almost all cases the LSMs could perform only as well as a multiple linear regression against SW, temperature and vapour pressure deficit (Fig 7). While an additional assessment of other outputs is required, the study highlighted

that there are likely systematic misrepresentations of simulated phenology and root-
water access in some of these models (Whitley et al., 2016). This is the first assessment
of its kind for investigating how well savanna dynamics are captured by modern TBMs,
and implies that without further development TBMs may have limited scope as
investigative tools for future projections of savanna ecosystems.

**5. Conclusion**
There is a large degree of uncertainty as to what impact climate change may have on the
structure and function of savanna ecosystems given their complex interaction with
climate. Because TBMs are the only interpreter of vegetation dynamics available to us
that can reconcile the combination of effects induced by climate change, their predictive
capability at representing savanna dynamics is of significant importance (Scheiter and
Higgins, 2009). For TBMs to have the necessary skill required to simulate savannas
under both present and future climate, model development must be concentrated
towards more adequate representations of phenology, root-water uptake, and
disturbance dynamics, notably fires. We outline our recommendations below in these
areas:
(1)    Phenology: A dynamic representation of how leaf area responds to seasonally

changing environment conditions, such that it becomes an emergent property of

the coupled dynamics of weather and ecosystem function.

(2)    Root-water uptake: Rooting depth and root distribution profiles that represent the

contrasting strategies of trees and seasonal grasses, including their temporal

dynamics. Additionally, root-water extraction schemes that can dynamically

respond to the wetting and drying of the soil over time, accessing soil-water from

where it is sustainably available rather than where the highest density of root

biomass occurs.

(3)    Disturbances: The role of disturbance (ubiquitous to all savannas) in keeping

savanna systems open needs to be accounted for in models. Models need to

represent the dynamic processes that capture the effect of fire on savanna

composition, particularly in suppressing woody growth. Additionally, recovery

periods whether through fire (re-sprouting) or cyclones (re-establishment)

860   should also be considered given the dynamic influence these events have on the

861   long-term carbon balance of savannas.

862 In addition to the recommended areas for TBM development above, we also stress that

863 any improvements made in the representation of the above processes must be followed

864 with a more complete evaluation and benchmarking of TBMs that considers multiple

865 data sources in order to better constrain model uncertainty. We have highlighted that

866 EC systems provide an unparalleled source of data for testing the predictive capability of

867 TBMs at simulating water and carbon exchange in savannas. The role of regional flux

868 communities, such as the OzFlux network (Baldocchi et al., 2001; Beringer et al., 2016a),

869 will be to advance applications of EC systems that target savanna characteristics

870 specifically. Indeed, more studies are needed that measure overstorey and understorey

871 turbulent fluxes (Moore et al., 2016b), given their ability to quantify the contribution of

872 co-dominant tree and grass functional types. Additionally, a greater use of ecological

873 transects as tools for model evaluation are needed to quantify the ability of TBMs to

874 simulate savanna behaviour over changing floristic structure and climate (Hutley et al.,

875 2011). However, additional ecological and physiological measurements are also needed

876 to test modelled representations of root-zone water dynamics, carbon allocation and

877 growth, phenology and the recovery of vegetation after major disturbance events (fire

878 and cyclones); dynamic processes that cannot be verified by EC datasets alone. Facilities

879 such as the Australian Super Site Network (Karan et al., 2016) run by the Terrestrial

880 Ecosystem Research Network (TERN) will be critical to the collection of

881 ecophysiological information that can inform how savanna dynamics are represented in

882 TBMs.

883 Finally, we outline that future model experiments and inter-comparison studies that

884 leverage EC and ecophysiological datasets should target each of the three previously

885 mentioned processes individually. These may include rooting depth and water

886 extraction experiments that test the sensitivity of TBMs to the dry season transition

887 period, or fire management studies that investigate how the floristic structure in TBMs

888 responds to variable fire frequency. Furthermore, such studies must also be conducted

889 for savanna sites that have well-established datasets to test the processes in question.

890 For example, we expect that any study that attempts to test or improve the

891 representation of fire dynamics in TBMs is to be conducted at a site that has a long-

892 running EC record (given the variable return time of fire events) and a full suite of

893 concurrent ecophysiological measurements that quantifies the response of vegetation

894 under post-fire recovery.


Current remote sensing observations suggest tree cover is increasing and grassland-
savanna-forest boundaries are changing (Bond, 2008) and these changes can have large
feedbacks to the earth-atmosphere system (Liu et al., 2015). There is still great
uncertainty in predicting the future of savanna biomes (Scheiter et al., 2015; Scheiter
and Higgins, 2009) and improving how savanna ecosystems are represented by TBMs
will likely encompass the consideration of additional processes that have not been
mentioned here.  This will no doubt include improved understanding of ecological
theory that will lead to improvements in modelling ecosystem demographics and tree-
grass interaction that will improve DGVMs. However, we believe that by identifying
these processes as the cause for degraded model performance in this ecosystem, a
roadmap for future development can be constructed that leverages the availability of
rich datasets and current state-of-knowledge.

**Acknowledgements**
This study was conducted as part of the 'Australian Savanna Landscapes: Past, Present
and Future' project funded by the Australian Research Council (DP130101566). The
support, collection and utilization of data were provided by the OzFlux network
(www.ozflux.org.au) and Terrestrial Ecosystem Research Network (TERN)
(www.tern.org.au), and funded by the ARC (DP0344744, DP0772981 and
DP130101566). PALS was partly funded by the TERN ecosystem Modelling and Scaling
infrAStructure (eMAST) facility under the National Collaborative Research
Infrastructure Strategy (NCRIS) 2013-2014 budget initiative of the Australian
Government Department of Industry. Rhys Whitley was supported through the ARC
Discovery Grant (DP130101566). Jason Beringer is funded under an ARC FT
(FT110100602). We acknowledge the support of the Australian Research Council Centre
of Excellence for Climate System Science (CE110001028). We thank Jason Beringer,
Caitlin Moore and Simon Scheiter for their permission to reproduce their results in this
study.

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

**Figure Captions:**

**Figure 1:** Global maps of (a) mean annual temperature and (b) mean annual rainfall for the period 1901 to 2015, determined from the CRU TS v. 3.23 dataset (Harris et al., 2014). The dataset has been clipped to the eco-floristic regions that approximate the global extent of savannas using the following plant functional types: tropical moist deciduous forest, tropical dry forest, subtropical dry forest and tropical shrubland (Ruesch and Gibbs, 2008).

**Figure 2:** Predicted changes to aboveground biomass over the period 2012 to 2100 for the Australian savanna region following three scenarios of projected rainfall seasonality according to IPCC SRES A1B (IPCC, 2007). The simulations were conducted using an adaptive Dynamic Global Vegetation Model (aDGVM) and predicts how (a) present day (2012) aboveground biomass changes, when (b) rainfall seasonality does not change, (c) rainfall seasonality increases, and (d) rainfall seasonality decreases over the forecast period. In all cases, the aboveground biomass of the Australian savanna region increases, with the magnitude of change determined by the degree of seasonality. Reprinted with permission from Scheiter et al. (2015).

**Figure 3:** Representation of how changes to (a) tree and grass phenology determines changes in (b) savanna gross primary productivity (GPP) for an Australian mesic savanna. Time-varying signals of tree and grass LAI (a) are determined from a MODIS bulk LAI product using the method of Donohue et al. (Donohue et al., 2009), and are prescribed as inputs to the Soil-Plant-Atmosphere (SPA) model to predict separate tree and grass GPP. Data and model outputs are from Whitley et al. (Whitley et al., 2016) (*this issue*).

**Figure 4:** Simulated differences in total ecosystem latent energy (LE) and the resultant evolution of soil moisture content through the soil profile over time for a mesic Australian savanna site. Simulations were conducted using two different terrestrial biosphere models (TBMs) that use different root-water extraction schemes. The top panel (a) shows outputs of savanna water flux using the Community Atmosphere Biosphere Land-surface Exchange (CABLE) model, where the density of the fine-root biomass weights soil-water extraction. The bottom panel (b) shows outputs of savanna water flux from the Soil-Plant-Atmosphere (SPA) model, where soil-water is dynamically extracted from where it sustainably available. Model outputs are from Whitley et al. (Whitley et al., 2015) (*this issue*).

**Figure 5:** The nonlinear response of net ecosystem productivity (NEP) as the canopy
regenerates after a fire event in 2003 at an Australian mesic savanna site. Fire
disturbance of a sufficient intensity suppresses productivity, pushing the savanna state
from sink to source over a period of 70 days at this site, as the rate of respiration
exceeds the rate of assimilation due to resprouting costs. Empirical models created
using an artificial neural network (NN) describe the 'UnBurnt' and 'Burnt' canopy NEP
responses over the same period, and their difference estimates the loss of canopy
productivity as a consequence of fire. Reprinted with permission from Beringer et al.
(Beringer et al., 2007).
**Figure 6:** Smoothed (10-day running mean) time-series of understorey (red),
overstorey (green) and total ecosystem (red) gross primary productivity (GPP) for a
mesic savanna site in northern Australia. Rainfall is represented as black bars. Negative
and positive signs represent the savanna state as a carbon source or sink respectively,
and orange arrows depict the occurrence of fire events. Data products for total
ecosystem and understorey GPP are inferred from observations of net ecosystem
exchange using eddy-covariance towers at heights of 23 m and 5 m respectively.
Overstorey GPP is determined as the difference between the ecosystem and the
understorey. Reprinted with permission from Moore et al. (Moore et al., 2016b) (*this*
*issue*).
**Figure 7:** Rank plot showing the average performance of 6 terrestrial biosphere models
(TBMs) across the North Australian Tropical Transect (NATT). The closer a model's rank
is to 1 the better its performance is at predicting latent energy (LE) and gross primary
productivity (GPP). Empirical benchmarks representing increasing levels of complexity
(emp1 < emp2 < emp3) are represented as grey lines, and coloured lines denote each
model. The lines have no scientific value and are used for visual purposes only.
Benchmarking and model evaluation data are from Whitley et al. (Whitley et al., 2015)
(*this issue*).

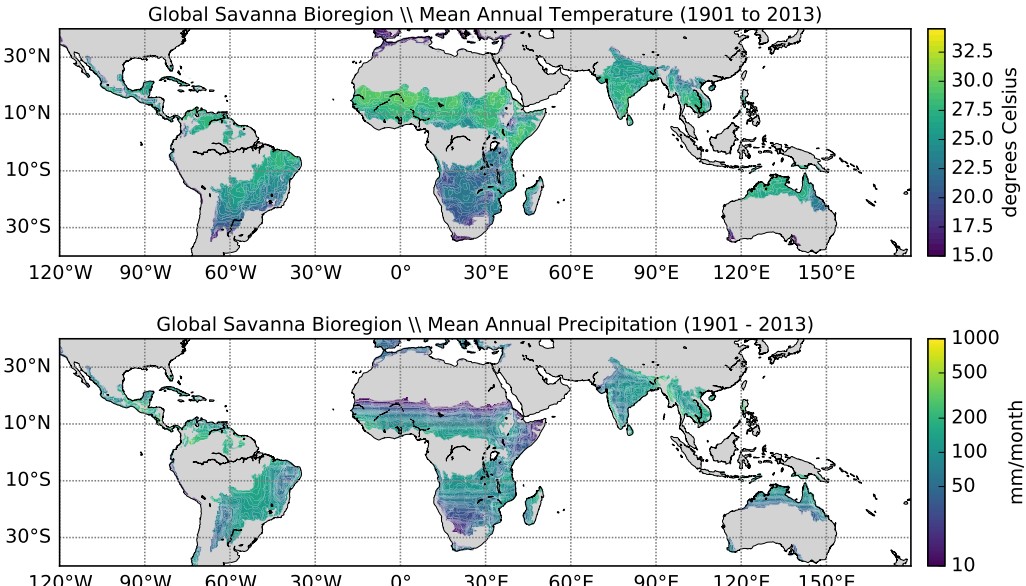

**Figure 1:** Global maps of (a) mean annual temperature and (b) mean annual rainfall for the period 1901 to 2015, determined from the CRU TS v. 3.23 dataset (Harris et al., 2014). The dataset has been clipped to the eco-floristic regions that approximate the global extent of savannas using the following plant functional types: tropical moist deciduous forest, tropical dry forest, subtropical dry forest and tropical shrubland (Ruesch and Gibbs, 2008).

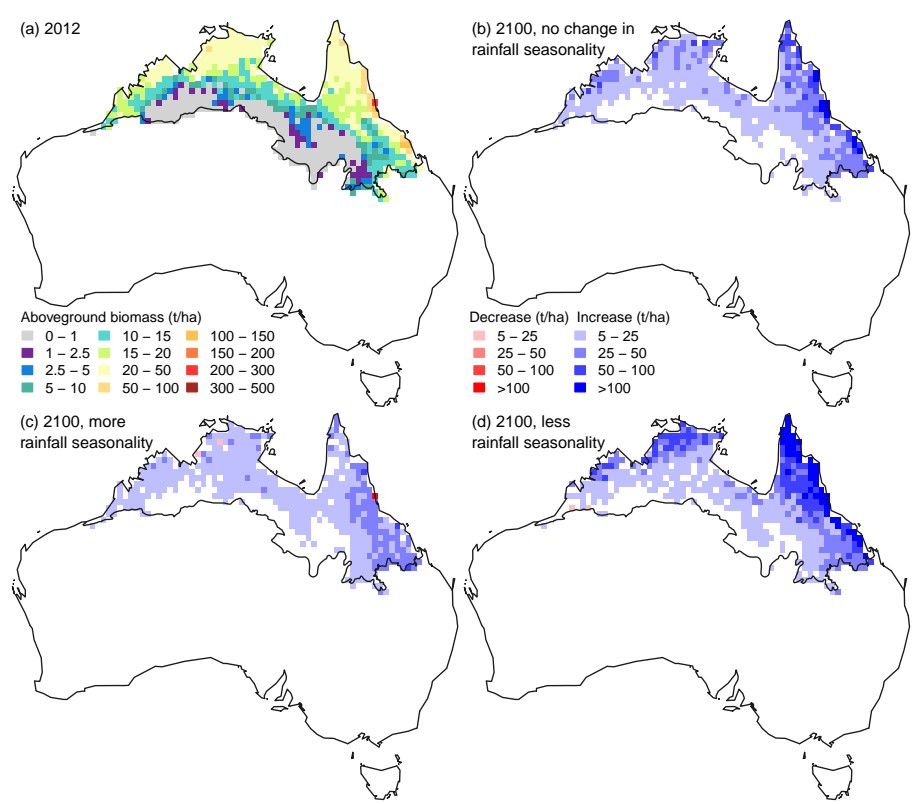

**Figure 2:** Predicted changes to aboveground biomass between 2012 and 2100 for the Australian savanna region following three scenarios of projected rainfall seasonality according to IPCC SRES A1B (IPCC, 2007). The simulations were conducted using an adaptive Dynamic Global Vegetation Model (aDGVM) shows predicted changes to (a) present day aboveground biomass, when (b) rainfall seasonality does not change, (c) rainfall seasonality increases, and (d) rainfall seasonality decreases. In all cases, the aboveground biomass of the Australian savanna region increases, with the magnitude of change determined by the degree of seasonality. Reprinted with permission from Scheiter et al. (2015).

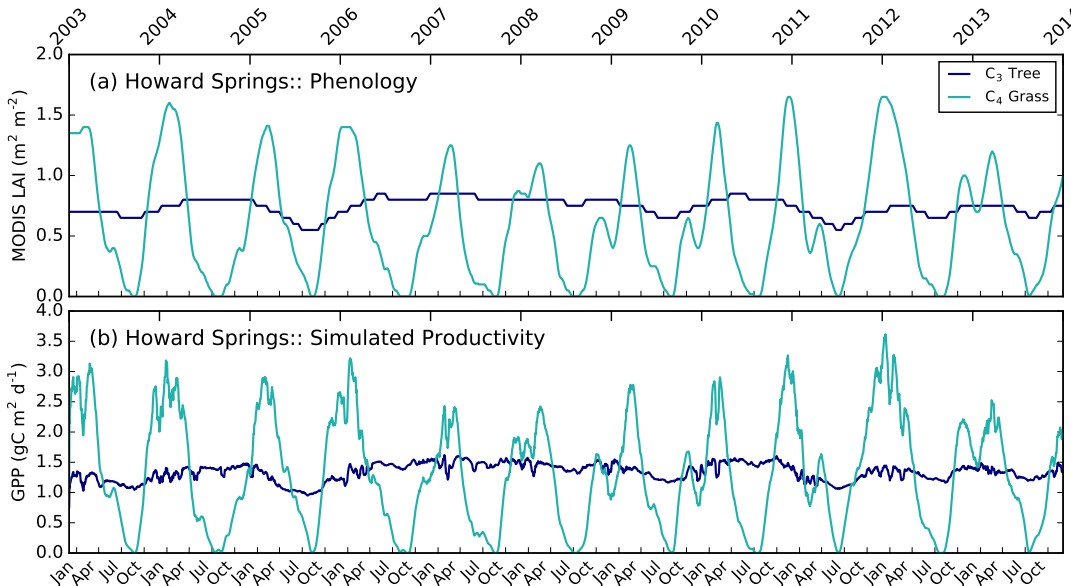

**Figure 3:** Representation of how changes to (a) tree and grass phenology determines changes in (b) savanna gross primary productivity (GPP) for an Australian mesic savanna. Time-varying signals of tree and grass LAI (a) are determined from a MODIS bulk LAI product using the method of Donohue et al. (2009), and are prescribed as inputs to the Soil-Plant-Atmosphere (SPA) model to predict separate tree and grass GPP. Data and model outputs are from Whitley et al. (2015) (*this issue*).

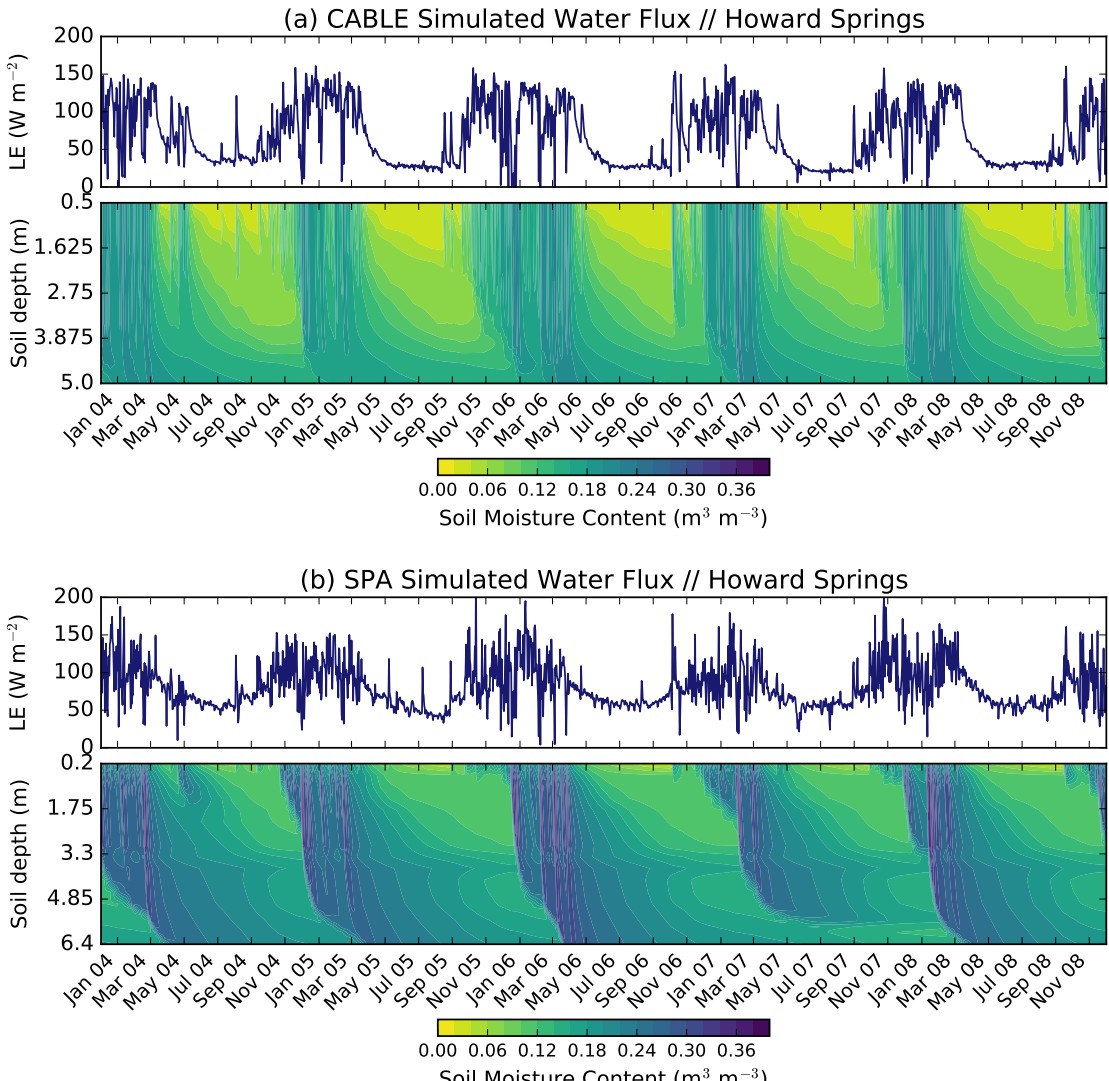

**Figure 4:** Simulated differences in total ecosystem latent energy (LE) and the resultant evolution of soil moisture content through the soil profile over time for a mesic Australian savanna site. Simulations were conducted using two different terrestrial biosphere models (TBMs) that use different root-water extraction schemes. The top panel (a) shows outputs of savanna water flux using the Community Atmosphere Biosphere Land-surface Exchange (CABLE) model, where the density of the fine-root biomass weights soil-water extraction. The bottom panel (b) shows outputs of savanna water flux from the Soil-Plant-Atmosphere (SPA) model, where soil-water is dynamically extracted from where it sustainably available. Model outputs are from Whitley et al. (2015) (*this issue*).

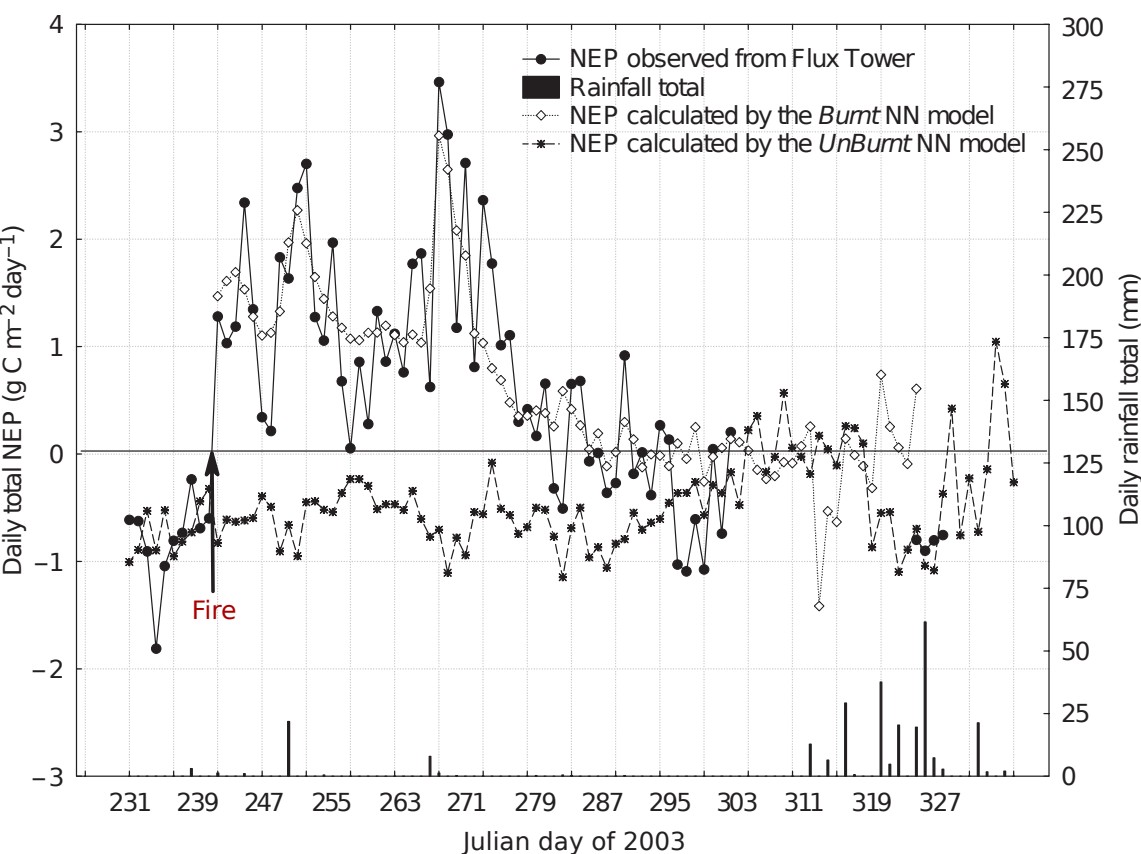

**Figure 5:** The nonlinear response of net ecosystem productivity (NEP) as the canopy regenerates after a fire event in 2003 at an Australian mesic savanna site. Fire disturbance of a sufficient intensity suppresses productivity, pushing the savanna state from sink to source over a period of 70 days at this site, as the rate of respiration exceeds the rate of assimilation due to resprouting costs. Empirical models created using an artificial neural network (NN) describe the 'UnBurnt' and 'Burnt' canopy NEP responses over the same period, and their difference estimates the loss of canopy productivity as a consequence of fire. Reprinted with permission from Beringer et al. (2007).

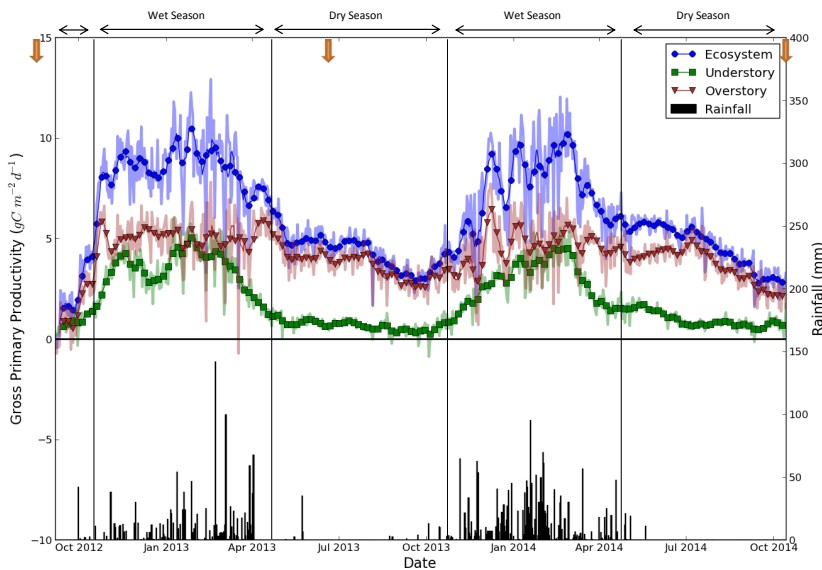

**Figure 6:** Smoothed (10-day running mean) time-series of understorey (red), overstorey (green) and total ecosystem (red) gross primary productivity (GPP) for a mesic savanna site in northern Australia. Rainfall is represented as black bars. Negative and positive signs represent the savanna state as a carbon source or sink respectively, and orange arrows depict the occurrence of fire events. Data products for total ecosystem and understorey GPP are determined from eddy-covariance towers at heights of 23 m and 5 m respectively. Overstorey GPP is determined as the difference between the ecosystem and the understorey. Reprinted with permission from Moore et al. (2016) (*this issue*).

**Figure 7:** Rank plot showing the average performance of 6 terrestrial biosphere models (TBMs) across the North Australian Tropical Transect (NATT). The closer a model's rank is to 1 the better its performance is at predicting latent energy (LE) and gross primary productivity (GPP). Empirical benchmarks representing increasing levels of complexity (emp1 < emp2 < emp3) are represented as grey lines, and coloured lines denote each model. The lines have no scientific value and are used for visual purposes only. Benchmarking and model evaluation data are from Whitley et al. (2015) (*this issue*).