# Peer review of "Challenges and opportunities in modelling savanna ecosystems"

_Biogeosciences, 2016_

## Referee Comment (RC1) · Anonymous Referee #1 · 6 Jun 2016

I enjoyed reading "Challenges and opportunities in modelling savanna ecosystems". I feel this paper provides a useful overview and is largely well-written. After the minor issues below are resolved I recommend this for publication. It will be a good addition to Biogeoscicenes.

L51: Remove "current-generation" as one might read this that previous generations were immune this challenge.

L60: Remove ",namely"

L67: Try "the effects"

L78: Something is off here as "and provide important in providing ecosystem services,..." makes no sense.

[Figure]

L84: Try "creates"

L88: The antecedent of "it" is unclear, use "fire" again here.

L96: I think you want "confounding" here?

L102: Try "the current generation of TBMs has..."

L126: I think you "proceed" given that you use present tense throughout here.

L184: Remove the first "region"

L189: Replace "For the" with "As an"

L190: Try "to emerge"

L207: Replace "...occupy the top ranks among terrestrial biomes, together contributing c. 30%" with "...contribute c. 30%"

L243: Try "are"

L246: "until"? Until what?

L268: Try "are"

L310: Try "partition" and "LAI."

L335: Remove comma after "advances". Also, I must state that the paper needs a good final proofreading. I have pointed out several (albeit minor) issues but have certainly not caught all the comma issues, and sundry other language faux pas.

L567: What is NATT? Maybe define in L560 above.

L572: Regarding your "as they cannot capture..." comment. I would dispute this especially as you invoke the space for time argument above. FLUXNET can quite do the same thing.

L575: Citations are off.

L591: I appreciate that the authors can't solve all these data limitations. But the "such data may be critical" comment is an interesting one, especially in the context of rather dear excavation studies. I'd like more detail. How many such excavation studies with what sampling design frame do you envision. That is, how do we move forward as a community to actually get the right data?

L603: In this section I would encourage the authors to cite some other developments here, e.g., ILAMB, that certainly hold promise to improve benchmarking. PALS is well and good but there is more afoot.

L692: Might NEON be a good idea? I must say I've noted a rather Australian-centric view of the literature. That is not bad, particularly in an OzFLux special issue, but again there are other things afoot and this is a review paper. And savannas do not exist solely in Australia.

L703: I am confused on the juxtaposition of long-term EC sites and fire return. A fire typically has adverse consequences for a FLUXNET installation. Are you advocating pre- and post-fire EC measurements?

---

## Referee Comment (RC2) · Anonymous Referee #2 · 9 Jun 2016

The topic of this review is interesting and highly relevant for improving and developing new ecological models for savanna ecosystem. It gives a detailed description of the complexity of savanna ecosystem dynamics by focusing on the main drivers which play a key role in controlling carbon and water fluxes of this ecosystem. The authors give clear recommendations on how should the ecological models be improved for having a better description of the savanna. I recommend publishing this review in Biogeosciences.

My minor comments are: (1) I found the description of additional dataset (ancillary and remote sensing data) which can be used to test the ecological model a little bit unclear. However, this part can be easily improved by the authors by adding more details on the variables which can be extrapolated from these datasets at the end of the session on "Datasets to inform model development" (P 18, L583). (2) A short discussion on the

scale mismatch between these datasets (including EC) and the model grid should be added in the "Model evaluation and benchmarking".

Specific comment

P5, L139: Please substitute "ecosystem types" with "ecoclimate regions"

P7, L183: Please define the two acronyms: LSM and DGVM

P9, L252: This should be section "3" and not section "2". Please check and re-number, where needed, all sections

P9, L252: Please eliminate ":" and the end of the title

P17, L524: Please eliminate "ground-based"

P17, L531-534: This sentence is not very clear for people that don't know very well how models and the eddy covariance system work. Please rephrase

P17, L534: I would like to use "ecosystem scale" instead of "spatial scale" to better give the idea of the spatial representativeness of EC data which are limited to the footprint area

P17, L534: Please refer to Aubinet et al., 2012

Aubinet, M., Vesala, T., and Papale, D.: Eddy Covariance – A Practical Guide to Measurement and Data Analysis, Springer, ISBN: 978-94-007-2351-1, 2012

P17, L535: Please add reference: Balzarolo et al., 2014; doi:10.5194/bg-11-2661-2014

P18, L567: Please explain NATT

P18, L583: Please revise this sentence, which is not very clear

P18, L573: Are you referring to long-term temporal predictions?

P21, L680: Please refer to Fluxnet (Baldocchi et al., 2001)

Baldocchi, D. D., Falge, E., Gu, L., Olson, R., Hollinger, D., Running, S., Anthoni, P., Bernhofer, C., Davis, K., Fuentes, J., Goldstein, A., Katul, G., Law, B., Lee, X., Malhi, Y., Meyers, T., Munger, J. W., Oechel, W., Pilegaard, K., Schmid, H. P., Valentini, R., Verma, S., Vesala, T., Wilson, K., and Wofsy, S.: FLUXNET: a new tool to study the temporal and spatial variability of ecosystem-scale carbon dioxide, water vapor and energy flux densities, B. Am. Meteorol. Soc., 82, 2415–2435, 2001.

P22, L693: Please also refer to NEON (National Ecological Observatory Network)

Figures

Figure 1: please add (a) and (b) in the figure and check the appropriate units to the y-axis (mean annual rainfall is correct in mm/month?)

Figure 2: it is not very clear the use of "2012" and "2011" in the legend. Please change or explain in the caption.

---

## Referee Comment (RC3) · Anonymous Referee #3 · 8 Jul 2016

This paper presents a helpful and informative review of some of the major challenges in quantifying and predicting structural and functional dynamics of savanna ecosystems with numerical models. The paper focuses on terrestrial biosphere models that mainly aim to predict water, energy, and carbon fluxes and balances as they interact with the atmosphere. This is a worthwhile focus but the title should probably be modified to reflect this specific focus. The manuscript has a lot of valuable content, and offers constructive advice regarding ways to improve TBM performance for savanna ecosystems. However the paper is less comprehensive than, I'd argue, it should be given its aims. Also the paper omits some important theory and themes in savanna ecology, and misrepresents some of the broad geographic context of global savannas. This review focuses on those elements and recommends revisions in those directions.

0) Regarding Root Water Uptake: This is a worthwhile focus for improvements of TBMs

but the authors have a rather specific read on the relevant literature. A few key citations I'd recommend are below, offering expanded perspectives on how to proceed with improving root water uptake in TBMs. Key considerations go beyond just prescribing rooting depth but also: dynamic uptake in response to soil water availability in the vertical profile, adaptive adjustments of the root distributions in response to water availability over seasonal and multi-year timscales, hydraulic redistribution along pressure gradients and via roots, soil water limitation function limiting productivity and evapotranspiration and associated water demand and water potential along the root to leaf and atmosphere continuum. I, too, caution against weighting root water uptake by fine root distribution because many plants are able to sustain water uptake and transpiration from deep taproots that access the saturated zone or deep unsaturated zone water sources even when fine roots (all concentrated near the surface) are in very dry soil layers. Furthermore, plant capacitance (storage) is important for accurately representing plant water potential, especially when water supply is limited or when root-to-leaf transport resistances inhibit water delivery to the site of transpiration at the leaf. Reinder A. Feddes, Holger Hoff, Michael Bruen, Todd Dawson, Patricia de Rosnay, Paul Dirmeyer, Robert B. Jackson, Pavel Kabat, Axel Kleidon, Allan Lilly, and Andrew J. Pitman, Modeling Root Water Uptake in Hydrological and Climate Models Bulletin of the American Meteorological Society 2001 82:12, 2797-2809. Lai, C.-T. & Katul, G. (2000) The dynamic role of root-water uptake in coupling potential to actual transpiration. Advances in Water Resources, 23, 427-439. Steudle, E., 2000. Water uptake by plant roots: an integration of views. Plant and Soil, 226(1), pp.45-56. Vrugt, J. A., M. T. vanWijk, J. W. Hopmans, and J. Šimunek (2001), One-, two-, and three-dimensional root water uptake functions for transient modeling, Water Resour. Res., 37(10), 2457–2470, doi:10.1029/2000WR000027.

1) A host of other processes of importance and interest in savannas are missed. For example, pulse response processes, stand-scale vegetation composition, plant-scale competitive interactions, stand-scale vegetation structure, landscape patterning of vegetation, nutrient cycling and interactions with herbivores, and more are all given little if any attention. Arguably many of those processes are important for representing savanna-atmosphere interactions, and for assessing savanna responses to global change factors. Given that this paper is intended to be a review of key processes that need to be considered to accurately model savanna ecological responses to global change factors, I would encourage additional discussion of these missed processes and their implications and importance for the stated aims.

2) Savanna ecologists would be underwhelmed by the three dynamic processes that are highlighted: phenology, root water uptake, and fire, given that these have long been the focus of their work going back many decades (e.g. Walter 1973). For example, the seminal work by Brian Walker (1981) is surprisingly absent from the present review even though this was foundational work identifying the importance of root zone separation and differential uptake zones for grass/herbaceous and woody PFTs in the savanna matrix. This was nicely tested in the Scholes and Walker (1993) book which is also missed. It is surprising that competitive interactions and differential resource access are not noted here, nor differential response of PFTs and species to single and multifactor drivers of CO2, drought, warming, and increasing VPD. While I agree that the three features highlighted in the present paper are essential and yet poorly represented (if at all) in TBMs, such models will still not be up to the task of predicting responses to global changes without representation of a host of other factors. Walter H (1973) Vegetation of the earth in relation to climate and the eco-physiological conditions. New York: Springer Scholes, R. J., and B. H. Walker (1993), An African Savanna: Synthesis of the Nylsvley Study, Cambridge Univ. Press, New York. Walker, B. H., D. Ludwig, C. S. Holling, and R. M. Peterman (1981), Stability of semi-arid savanna grazing systems, J. Ecol., 69, 473– 498.

3) Grazing and browsing are of central importance in many of the world's savannas, strongly influencing vegetation cover, loss of productivity and biomass, species composition, and affecting site fertility but this driver is hardly mentioned, receiving just one or two sentences. A bit more on this subject would seem warranted for such a review.

4) Corresponding to the above, a nod to the alternate stable states literature is missing here, including Walker '81, Noy-Meir 1975, and others mentioned below. Noy-Meir I (1975) Stability of grazing systems: an application of predator prey graphs. Journal of Ecology, 63, 459–481. Jeltsch F, Weber GE, Grimm V (2000) Ecological buffering mechansims in savannas: a unifying theory of long-term tree-grass coexistence. Plant Ecology, 161, 161–171. Scheffer M, Carpenter SR (2003) Catastrophic regime shifts in ecosystems: linking theory to observation. Trends in Ecology & Evolution, 18, 648–656. van de Koppel J, Rietkerk M, Weissing FJ (1997) Catastrophic vegetation shifts and soil degradation in terrestrial grazing systems. Trends in Ecology & Evolution, 12, 352–356. Westoby M, Walker B, Noy-Meir I (1989) Range management on the basis of a model which does not seek to establish equilibrium. Journal of Arid Environments, 17, 235–239. Williams, C.A. & Albertson, J.D. (2006) Dynamical effects of the statistical structure of annual rainfall on dryland vegetation. Global Change Biology, 12, 1-16.

5) Discussion of the global context and diversity of savanna attributes and strategies is lacking and in some ways misleading. Section 2.1, particularly Line 169+: The language here misrepresents the growth and longevity strategies of woody plants in Africa. Many of the woody species in at least southern Africa do indeed have deep roots but groundwater is deep (probably deeper than in much of Australia) so there is less potential to rely on near surface (<10 m) water sources. The Archibald and Scholes '97 paper does not mention roots once, and says nothing about strategies of water access. The Higgins '11 paper also offers little on root water uptake. Both of those papers do indeed discuss and quantitatively document phenological dynamics, but neither indicates that the full woody component of southern African savannas is deciduous (indeed Acacia sp. often retain leaves consistently through the dry season). However both represent only southern African ecosystems at best (really, Kruger Park). Yet this statement is as grandiose as generalizing from these studies to all African and South American savannas! That's stretching it a bit, no? A much broader literature must be invoked if the authors truly want to discuss geographic patterns of root water uptake, and diversity in savanna traits and properties. Furthermore, this must consider not just phenology

but also water availability in the unsaturated and saturated zones, and not confuse mesic and arid savanna types. The present interpretation seems to conflate shallow groundwater availability or its absence with a difference in plant strategy. However, woody species of savannas around the world "favour a long-term strategy of conservative growth that is insured against an unpredictable climate", not just those in Australia. To include more on the global biogeography of savannas relevant to a modeling context I'd recommend some additional reading (and citation of) works in: Hill, Michael J. and Hanan, Niall P. eds (2011). Ecosystem Function in Savannas: Measurement and Modeling at Landscape to Global Scales. (CRC Press, Boca Raton, Florida) 559 pp.

6) The Pulse-Reserve paradigm in dryland ecology is noticeably absent from this review despite the well-known importance of rainfall pulses in organizing complex ecological and biophysical dynamics in water-limited environments. Many plant and ecosystem phenological dynamics are organized around rainfall pulses, including leaf-out and senescence, up- and down-regulation of productivity, respiration and decomposition bursts, reproduction and establishment events, and so forth. "Pulse" is not mentioned once in the current review. Chesson P, Gebauer RLE, Schwinning S et al. (2004) Resource pulses, species interactions, and diversity maintenance in arid and semi-arid environments. Oecologia, 141, 236–253. Huxman, T.E., Snyder, K.A., Tissue, D., Leffler, A.J., Ogle, K., Pockman, W.T., Sandquist, D.R., Potts, D.L. & Schwinning, S. (2004) Precipitation pulses and carbon fluxes in semiarid and arid ecosystems. Oecologia, 141, 254-268. Jenerette, G.D., Scott, R.L. & Huxman, T.E. (2008) Whole ecosystem metabolic pulses following precipitation events. Functional Ecology, 22, 924-930. Noy-Meir, I. (1973), Desert ecosystems: Environment and producers, Annu. Rev. Ecol. Syst., 4, 25– 44. Ogle, K. & Reynolds, J.F. (2004) Plant responses to precipitation in desert ecosystems: integrating functional types, pulses, thresholds, and delays. Oecologia, 141, 282-294. Williams, C.A. & Albertson, J.D. (2004) Soil moisture controls on canopy-scale water and carbon fluxes in an African savanna. Water Resources Research, 40, 1-14. Williams, C.A., Hanan, N.P., Scholes, R.J. & Kutsch, W. (2009) Complexity in water and carbon dioxide fluxes following rain pulses in an African savanna. Oecologia,

7) Section 3.1: Possibly also mention potential for additional measurements to inform root water uptake dynamics (maybe around L590): -experimental use of isotopes to trace root water uptake dynamics (see work of Todd Dawson's lab for example). -standard field-measured sapflow and leaf gas exchange are surprisingly not mentioned but can be particularly useful when coupled with detailed soil moisture profile measurements, where changes over time directly indicate the effects of water uptake. -weighing lysimeter studies, while very intensive, have also been used to detect whole plant uptake. -groundwater wells would also be enormously helpful and are so often missed in ecological and even hydrological studies in savannas (and other ecosystems), yet are critical for characterizing the availability and dynamics of deep water sources. -groundwater maps, where available, are low hanging fruit for incorporation into spatial applications of TBMs. -another key thing that is missing is detailed mapping of C3 and C4 vegetation types (grasses/herbaceous), and their separate phenologies. -remotely sensed surface temperature is another valuable constraint on ecosystem water status (I think Damian Bonal was working on this and published on it).

8) Conclusions go uncomfortably beyond what is supported in this paper and stray from the paper's clear focus on how to improve TBM performance for savannas. For example: "Projected higher temperatures and rainfall variability, potentially promoting 645 more frequent fires, could favour C4 grasses in mesic savanna, while drier conditions are 646 expected to increase tree mortality in semi-arid savanna. Conversely, increases to 647 atmospheric $CO_2$ are expected to favour C3 trees, reflecting woody encroachment that is 648 already observed in many savannas globally (Donohue et al., 2009). Climate change 649 therefore has the potential to alter the carbon balance, which may have major feedbacks 650 on global climate and biogeochemical cycling."

9) Again, it is recommended that the authors expand the scope of highlights to also emphasize ecosystem structural and compositional dynamics that are of central importance to TBM processes: particularly differential resource acquisition (primarily water)

and competitive interactions. E.g. around L694... model and data efforts should also target those attributes of savannas. Perhaps the authors roll all of that into "phenology" but I'd argue that this is a mistake, where phenology is only one component of vegetation dynamics. The underlying competitive interactions, mortality and growth dynamics, and how these shift in response to a suite of climate, atmospheric compositional, soil fertility, land use and other global change factors could receive more attention in this review.

Some Details:

Why is root-water hyphenated? Do you mean ground-water or soil-water? Probably just drop the hyphen throughout.

Line 69+: not just "environmental conditions" but also biophysical and ecological conditions... that is, the ecosystem properties are themselves changing and this must be represented.

Line 96: "confronting task" reword, unclear

L100: "underperformed for savanna ecosystems" is too vague... what, specifically, lacks accuracy? "under" relative to what, other PFT or biome types, compared to data?

L105: "physical [and biological]"... most of these are not physical parameters.

---

## Author Comment (AC1) · 15 Aug 2016

**Author responses to reviewer and editor comments for manuscript submission BG-2016-190**

Below are outlined our responses to the comments from the two anonymous reviewers for our paper entitled: "**Challenges and opportunities in modelling savanna ecosystems.**"

We thank all three reviewers for their valuable feedback and insights. Both reviewers #1 and #2 provided only minor corrections, which we are largely in agreement with and have consequently adjusted the text of the paper to reflect this. Regarding reviewer #3, while their extensive commentary and insight on savanna ecology is highly informative and greatly appreciated, we feel that many suggested changes fall outside the scope of the paper's core messages. The introductory sections (1 and 2) have attempted to cover the nature of savanna and many of the features that differentiate them globally. This paper builds on previous work (accepted for publication in this special issue: Whitley et al. (2016)) that highlighted a set of processes related to savanna dynamics that are currently deficient in TBMs. In this paper, we have therefore set out to focus our discussion on how these processes are currently misrepresented (or absent) in TBMs and offer recommendations on how they could be developed to improve the predictive capability of such models in simulating the turbulent fluxes of savannas.

We have therefore primarily focused discussion on three dynamic processes: i) phenology, ii) root water uptake and iii) disturbance (particularly fire), which are the first-order controls on savanna water and carbon exchange and should therefore be critical areas of future model development. Reviewer #3 raises issues regarding the lack of discussion relating to ecological processes such as tree-grass demographics, canopy structure, pulse responses to rainfall, etc. However, they could be seen as emergent behaviour resulting from the dynamical processes highlighted in this paper, and an expanded discussion of these issues could distract from what we consider the primary deficiencies that TBMs currently face in simulating savanna ecosystems. Furthermore, not all TBMs have the capability (or goal) of simulating complex vegetation dynamics. This is not to say these ecological properties of savannas are not important, on the contrary we believe that TBMs need to be able to replicate these effects, but this would be a consequent step after the first order processes we have highlighted have been improved. Nevertheless, we have attempted to include some of reviewer #3's suggestions where we feel they were appropriate and within the scope of the paper's message. We also wish to stress that reviewer #3 has raised important issues

ubiquitous within savanna ecology that could serve as a basis for future work and would serve as natural progression to what we have presented here.

Reviewer comments are numbered below, where we have answered each to the best of our ability and made the appropriate changes where necessary in our manuscript. Once again, we would like to thank the reviewers and the editor for taking the time to examine this work and provide valuable feedback.

| Anonymous Reviewer #1 |
| --- |

**L51:** Remove "current-generation" as one might read this that previous generations were immune this challenge.

**Author response:** Done.

**L60:** Remove ",namely"

**Author response:** Done.

**L67:** Try "the effects"

**Author response:** Done.

**L78:** Something is off here as "and provide important in providing ecosystem services,..." makes no sense.

**Author response:** Has been corrected to: "… and **are** important in providing ecosystem services…"

**L84:** Try "creates"

**Author response:** "and create demographic …" has been changed to "that create demographic…"

**L88:** The antecedent of "it" is unclear, use "fire" again here.

**Author response:** Done

**L96:** I think you want "confounding" here?

**Author response:** Done

**L102:** Try "the current generation of TBMs has..."

**Author response:** Done

**L126:** I think you "proceed" given that you use present tense throughout here.

**Author response:** Done

**L184:** Remove the first "region"

**Author response:** Done

**L189:** Replace "For the" with "As an"

**Author response:** Done

**L190:** Try "to emerge"

**Author response:** Done

**L207:** Replace "...occupy the top ranks among terrestrial biomes, together contributing c. 30%" with "...contribute c. 30%"

**Author response:** Done

**L243:** Try "are"

**Author response:** Done

**L246:** "until"? Until what?

**Author response:** Sentence is complete now: "…until *later in this* paper."

**L268:** Try "are"

**Author response:** Done

**L310:** Try "partition" and "LAI."

**Author response:** Done

**L335:** Remove comma after "advances". Also, I must state that the paper needs a good final proofreading. I have pointed out several (albeit minor) issues but have certainly not caught all the comma issues, and sundry other language faux pas.

**Author response:** Done, and furthermore we have gone through the text again to identify other typos and grammatical issues.

**L567:** What is NATT? Maybe define in L560 above.

**Author response:** The acronym has now been correctly added after the definition given at the beginning of the paragraph.

**L572:** Regarding your "as they cannot capture. . ." comment. I would dispute this especially as you invoke the space for time argument above. FLUXNET can quite do the same thing.

**Author response:** The reviewer is quite correct, and we have qualified this statement to say: "as they cannot *completely* capture. . ."

**L575:** Citations are off.

**Author response:** Fixed

**L591:** I appreciate that the authors can't solve all these data limitations. But the "such data may be critical" comment is an interesting one, especially in the context of rather dear excavation studies. I'd like more detail. How many such excavation studies with what sampling design frame do you envision? That is, how do we move forward as a community to actually get the right data?

**Author response:** Root excavation studies as mentioned was only given here in a general sense as an example, however we see the reviewers point that such field campaigns are complex and expensive (in cost, time and labour), such that this warrants a further expansion of detail. We have therefore added the following lines to qualify our statement not just for root excavations, but also for all ecological trait information:

*"We recommend that future EC studies, particularly along transects as mentioned above, should include intensive field campaigns that are targeted towards a more complete characterisation of the site. This would include key flux measurements (e.g. sapflow, stomatal conductance, leaf water potentials, deep soil water measurements,  root excavations and the collection of plant trait data (e.g, leaf mass per area, capacitance, Rubisco activity, etc.) within the footprint of an EC tower. Collaborative research networks,*

*such as TERN (Terrestrial Ecosystem Research Network), NEON (National Ecological Observatory Network) and SAEON (South African Environmental Observation Network) that have the resources and infrastructure to conduct such campaigns will be instrumental to meet these demands for more observational data."*

**L603:** In this section I would encourage the authors to cite some other developments here, e.g., ILAMB, that certainly hold promise to improve benchmarking. PALS is well and good but there is more afoot.

**Author response:** We agree with the reviewer and have incorporated mention of ILAMB and other model intercomparison projects (e.g. PILPS, C4MIP) into this part of the discussion.

**L692:** Might NEON be a good idea? I must say I've noted a rather Australian-centric view of the literature. That is not bad, particularly in an OzFLux special issue, but again there are other things afoot and this is a review paper. And savannas do not exist solely in Australia.

**Author response:** We have now included mention of NEON as well as SAEON at the end of Section 3.1 (see response to comment on L591)

**L703:** I am confused on the juxtaposition of long-term EC sites and fire return. A fire typically has adverse consequences for a FLUXNET installation. Are you advocating pre- and post-fire EC measurements?

**Author response:** Fire is a a frequent occurrence in savanna and has a major impact on fluxes and we propose that savanna FLUXNET installations quantify the effects of fire as per Beringer et al. (2007) and as was mentioned in Section 2.3. In this regard we are advocating pre- and post-fire measurements, as this would allow TBMs (those that include the simulation of fire) to be tested on whether they have the capability to simulate the nonlinear response of the canopy (due to scorching and reduced surface albedo) during the post-fire recovery period.

My minor comments are:

**Comment 1:** I found the description of additional dataset (ancillary and remote sensing data) which can be used to test the ecological model a little bit unclear. However, this part can be easily improved by the authors by adding more details on the **variables** which can be extrapolated from these datasets at the end of the session on "Datasets to inform model development" (P18, L583).

**Author response:** We have now made reference to common model parameters that would benefit from the collection of specific environmental information. This text is quoted below as:

*"Digital soil atlases also provide an excellent resource in parameterising simulated soil profiles (e.g. Isbell, 2002; Sanchez et al., 2009). However, the spatial resolution of these data products can be coarser than the operating resolution of many TBMs, such that site-level measurements should be used when possible. Excavation studies that quantify savanna tree root-systems (Chen et al., 2004) and soil-moisture probes installed at greater depths (> 2 m) are informative about the evolution of the soil-root zone over time (e.g. surface root density, root depth), and such data may be critical to understanding whether current root-water extraction schemes in TBMs are capable of simulating the dry season response of savanna tree species (Whitley et al., 2015). Finally, localised observations of plant traits such leaf-mass per area, leaf capacitance, tree height, etc. are needed to inform a better parameterisation of savanna specific PFTs (Cernusak et al., 2011). For example, specific leaf-level information such as Rubisco activity ($V_{cmax}$) and RuPB regeneration ($J_{max}$) for both $C_3$ and $C_4$ plants are critically needed to inform the Farquhar leaf photosynthesis models (Farquhar et al., 1980), while information on $g_s$ and leaf water potential ($\Psi_{leaf}$) are important in parameterising the many stomatal conductance models used in TBMs (Ball et al., 1987; Medlyn et al., 2011; Williams et al., 1996). Leaf capacitance and water potential data are also critically important in characterising model sensitivity to drought (Williams et al., 2001), but this information is severely lacking for savannas."*

**Comment 2:** A short discussion on the scale mismatch between these datasets (including EC) and the model grid should be added in the "Model evaluation and benchmarking".

**Author response:** We agree that the intention of almost all TBMs is to be run at the global scale does not match the scale at which validation occurs. Model evaluation of TBMs occurs at the ecosystem scale (a moderate resolution of ~1 km), commensurate with resolution of flux tower data and remotely sensed data products (e.g. Best et al., 2015; Blyth et al., 2010). Consequently, there is little scale mismatch between what is used to run (inputs) and validate (outputs) the models.

**Specific comments**

**P5, L139:** Please substitute "ecosystem types" with "ecoclimate regions"

**Author response:** Done

**P7, L183:** Please define the two acronyms: LSM and DGVM

**Author response:** Done

**P9, L252:** This should be section "3" and not section "2". Please check and re-number, where needed, all sections

**Author response:** Done. All section numbering has been updated accordingly.

**P9, L252:** Please eliminate ":" and the end of the title

**Author response:** Done

**P17, L524:** Please eliminate "ground-based"

**Author response:** Done

**P17, L531-534:** This sentence is not very clear for people that don't know very well how models and the eddy covariance system work. Please rephrase

**Author response:** We have changed the sentence in accordance with this request:

*"Turbulent fluxes measured by EC systems that include net ecosystem exchange and latent and sensible heat are common model outputs, such that this information is commonly used to validate TBMs. Local meteorological forcing (e.g. short-wave irradiance; SW, air temperature, rainfall, etc.) that is concurrently measured with the turbulent fluxes by other instruments (rainfall and temperature gauges, radiation sensors, etc.) are common model inputs and are used to drive TBMs. "*

**P17, L534:** I would like to use "ecosystem scale" instead of "spatial scale" to better give the idea of the spatial representativeness of EC data which are limited to the footprint area

**Author response:** Done

**P17, L534:** Please refer to Aubinet et al., 2012

Aubinet, M., Vesala, T., and Papale, D.: Eddy Covariance – A Practical Guide to Measurement and Data Analysis, Springer, ISBN: 978-94-007-2351-1, 2012

**Author response:** Done

**P17, L535:** Please add reference: Balzarolo et al., 2014; doi:10.5194/bg-11-2661-2014

**Author response:** Done

**P18, L567:** Please explain NATT P18, L583: Please revise this sentence, which is not very

clear

**Author response:** Definition has now been inserted, and sentence now reads as:

*"A recent model intercomparison study by Whitley et al. (2015) used turbulent flux observations sampled along the NATT to evaluate a set of six TBMs, and documented only poor to moderate performance for those savanna sites."*

**P18, L573:** Are you referring to long-term temporal predictions?

**Author response:** No, what we are alluding to in this sentence is that eddy-covariance systems may not have been running for long enough (i.e. the sampling period of the time-series) at a site to pick up any long-term structural changes (e.g. such as those caused by cyclones, or large fires). If a model is attempting to simulate a demographic shift in vegetation (i.e. the tree/grass ratio) then longer term datasets such as those from satellite would be more useful in validating this prediction.

**P21, L680:** Please refer to Fluxnet (Baldocchi et al., 2001)

Baldocchi, D. D., Falge, E., Gu, L., Olson, R., Hollinger, D., Running, S., Anthoni, P., Bernhofer, C., Davis, K., Fuentes, J., Goldstein, A., Katul, G., Law, B., Lee, X., Malhi, Y., Meyers, T., Munger, J. W., Oechel, W., Pilegaard, K., Schmid, H. P., Valentini, R., Verma, S., Vesala, T., Wilson, K., and Wofsy, S.: FLUXNET: a new tool to study the temporal and spatial variability of ecosystem-scale carbon dioxide, water vapor and energy flux densities, B. Am. Meteorol. Soc., 82, 2415–2435, 2001.

**Author response:** Done

**P22, L693:** Please also refer to NEON (National Ecological Observatory Network)

**Author response:** Done

**Figure 1:** please add (a) and (b) in the figure and check the appropriate units to the y-axis (mean annual rainfall is correct in mm/month?)

**Author response:** Done, the units are correct but we have rescaled the values to be mm/year.

**Figure 2:** it is not very clear the use of "2012" and "2011" in the legend. Please change or explain in the caption.

**Author response:** Done.

**Anonymous Reviewer #3**

*Note: we have removed the listing of references for brevity, but can be referred to in the reviewers original commentary.*

**Comment 0:** Regarding Root Water Uptake: This is a worthwhile focus for improvements of TBMs but the authors have a rather specific read on the relevant literature. A few key citations I'd recommend are below, offering expanded perspectives on how to proceed with improving root water uptake in TBMs. Key considerations go beyond just prescribing rooting depth but also: dynamic uptake in response to soil water availability in the vertical profile, adaptive adjustments of the root distributions in response to water availability over seasonal and multi-year timescales, hydraulic redistribution along pressure gradients and via roots, soil water limitation function limiting productivity and evapotranspiration and associated water demand and water potential along the root to leaf and atmosphere continuum. I, too, caution against weighting root water up- take by fine root distribution because many plants are able to sustain water uptake and transpiration from deep taproots that access the saturated zone or deep unsaturated zone water sources even when fine roots (all concentrated near the surface) are in very dry soil layers. Furthermore, plant capacitance (storage) is important for accurately representing plant water potential, especially when water supply is limited or when root-to-leaf transport resistances inhibit water delivery to the site of transpiration at the leaf.

**Author response:** We agree that other belowground processes the reviewer has mentioned are critical to simulating integrated representation of the soil-vegetation-atmospheric continuum (SVAC). However, we believe that Section 2.2 on root-water access and uptake does cover most of the processes the reviewer has mentioned, particularly dynamic uptake. The aim of this section is to argue for a better representation of root water access in and around the rhizosphere, as this is describes the supply end of the SVAC. Consequently it acts as a first order control for savanna water and carbon exchange in the dry season, which has not been modelled well in past studies. Hence we have specifically focused our discussion on this limitation, and that the modelling community could easily address this. Regarding plant capacitance, we agree that this is a critical parameter towards vegetation drought responses and have amended the second sentence of the first paragraph of Section 2.2 to reflect this.

*"For seasonally dry climates (a fundamental characteristic of savanna ecoclimatic regions), productivity is primarily limited by dry season water-availability (Kanniah et al., 2010), which is largely determined by plant regulation of water transport (through leaf stomatal conductance and stem capacitance) and the root zone water storage capacity and access (distribution of fine root biomass)."*

We appreciate the reviewer's contribution of further references to this section and have placed these appropriately within the text where possible.

**Comment 1:** A host of other processes of importance and interest in savannas are missed. For example, pulse response processes, stand-scale vegetation composition, plant-scale competitive interactions, stand-scale vegetation structure, landscape patterning of vegetation, nutrient cycling and interactions with herbivores, and more are all given little if any attention. Arguably many of those processes are important for representing savanna-atmosphere interactions, and for assessing savanna responses to global change factors. Given that this paper is intended to be a review of key processes that need to be considered to accurately model savanna ecological responses to global change factors, I would encourage additional discussion of these missed processes and their implications and importance for the stated aims.

**Author response:** We agree that these are important processes in savanna ecology, however *"stand-scale vegetation composition, plant-scale competitive interactions, stand-scale vegetation structure, etc."* could be seen as emergent properties of the three dynamic

processes that our paper focuses on. Given that the discussion is focussed on the first order controls on savanna water and carbon exchange and how these are likely misrepresented (or absent) from modern TBMs, further discussion about such ecological processes is outside the scope of the paper. In order for TBMs to better simulate savanna intra- and interspecies competition, canopy structure, etc. the 3 first-order dynamic processes we have mentioned here must be addressed before all else.

**Comment 2:** Savanna ecologists would be underwhelmed by the three dynamic processes that are highlighted: phenology, root water uptake, and fire, given that these have long been the focus of their work going back many decades (e.g. Walter 1973). For example, the seminal work by Brian Walker (1981) is surprisingly absent from the present review even though this was foundational work identifying the importance of root zone separation and differential uptake zones for grass/herbaceous and woody PFTs in the savanna matrix. This was nicely tested in the Scholes and Walker (1993) book which is also missed. It is surprising that competitive interactions and differential resource access are not noted here, nor differential response of PFTs and species to single and multifactor drivers of CO2, drought, warming, and increasing VPD. While I agree that the three features highlighted in the present paper are essential and yet poorly represented (if at all) in TBMs, such models will still not be up to the task of predicting responses to global changes without representation of a host of other factors.

**Author response:** We are fully aware of the decades of ecological research on the three dynamic processes that are highlighted in our review section; phenology, root water uptake, and fire, but these have been poorly captured in TBMs to date, and this is the focus of this paper. Our goal is not to provide a review of determinants of savanna structure and function but to target the failing of TBMs. Again we refer back to our response to Comment 2 that this type of discussion is outside the scope of the paper. Additionally, we are reluctant to argue for the inclusion of tree-grass root niche separation as a missing critical factor, primarily as there is limited empirical evidence (that we are aware of) to support the hypothesis that it is important in savannas. Indeed, studies such as that of February et al. (2010) have shown both tree and grass roots to be co-occurring at the same soil depth for an African savanna, as has been documented in Australian savanna (Chen et al. 2004, Eamus et al. 2003)

**Comment 3:** Grazing and browsing are of central importance in many of the world's savannas, strongly influencing vegetation cover, loss of productivity and biomass, species com- position, and affecting site fertility but this driver is hardly mentioned, receiving just one or two sentences. A bit more on this subject would seem warranted for such a review.

**Author response:** We certainly agree that grazing and browsing are one of many important factors that modulate the savanna structure and state. However, animals are not a modelled process in TBMs and this paper is focussed on simulated processes that control savanna water and carbon exchange.

**Comment 4:** Corresponding to the above, a nod to the alternate stable states literature is missing here, including Walker '81, Noy-Meir 1975, and others mentioned below.

**Author response:** We believe this has been covered in the sections on the *Implications of Climate Change* and *Phenology.*

**Comment 5:** Discussion of the global context and diversity of savanna attributes and strategies is lacking and in some ways misleading. Section 2.1, particularly Line 169+: The language here misrepresents the growth and longevity strategies of woody plants in Africa. Many of the woody species in at least southern Africa do indeed have deep roots but groundwater is deep (probably deeper than in much of Australia) so there is less potential to rely on near surface (<10 m) water sources. The Archibald and Scholes '97 paper does not mention roots once, and says nothing about strategies of water access. The Higgins '11 paper also offers little on root water uptake. Both of those papers do indeed discuss and quantitatively document phenological dynamics, but neither indicates that the full woody component of southern African savannas is deciduous (indeed Acacia sp. often retain leaves consistently through the dry season). However both represent only southern African ecosystems at best (really, Kruger Park). Yet this statement is as grandiose as generalizing from these studies to all African and South American savannas! That's stretching it a bit, no? A much broader literature must be invoked if the authors truly want to discuss geographic patterns of root water uptake, and diversity in savanna traits and properties. Furthermore,

this must consider not just phenology but also water availability in the unsaturated and saturated zones, and not confuse mesic and arid savanna types. The present interpretation seems to conflate shallow groundwater availability or its absence with a difference in plant strategy. However, woody species of savannas around the world "favour a long-term strategy of conservative growth that is insured against an unpredictable climate", not just those in Australia. To include more on the global biogeography of savannas relevant to a modeling context I'd recommend some additional reading (and citation of) works in: Hill, Michael J. and Hanan, Niall P. eds (2011). Ecosystem Function in Savannas: Measurement and Modeling at Landscape to Global Scales. (CRC Press, Boca Raton, Florida) 559 pp.

**Author response:** We have included citations from Bowman and Prior (2005), Lehmann et al. (2011, 2014) and Stevens et al. (2016) that support our argument that woody species among continents can be clustered as we have stated in the text. The Archibald and Scholes (1997) is a mistake and should be Scholes and Archer (1997), now corrected, while Higgins et al. (2011) has been removed given that it does not directly support our claims. We do agree that not all African woody species are shallow-rooted and have qualified the statement to read as:

"*Canopies of the African and South American savanna regions are predominantly characterised by deciduous woody species that are in most cases (although not always) shallow-rooted and follow a short-term growth strategy that maximises productivity while environmental conditions are favourable*"

We do not believe that expanding the discussion on global savanna attributes to cover every degree of separation of plant traits between and within continents benefits our argument on the 3 dynamics processes that we believe currently hamper TBM performance in savanna ecosystems. The primary point of this section is to highlight and provide context that these differences suggest that savannas cannot be lumped into some generalised group or plant functional type (PFT), but show clear distinctions. Rather, we argue that region specific PFTs will likely be required for good model performance in this ecosystem. However, this is not to say that the assignment of correct trait information is the answer to improved model performance in savannas, rather a better representation of phenology, root-water uptake and disturbance (fire) in TBMs is required to fill this gap.

**Comment 6:** The Pulse-Reserve paradigm in dryland ecology is noticeably absent from this

review despite the well-known importance of rainfall pulses in organizing complex ecological and biophysical dynamics in water-limited environments. Many plant and ecosystem phenological dynamics are organized around rainfall pulses, including leaf-out and senescence, up- and down-regulation of productivity, respiration and decomposition bursts, reproduction and establishment events, and so forth. "Pulse" is not mentioned once in the current review.

**Author response:** We agree that the ecosystem response to rainfall pulses in arid climates is an important behaviour for TBMs to capture. However, TBMs should have the capability to capture this response, and if not, it is likely not a missing mechanism per se, rather a lack of sensitivity. We have discussed issues related to this response in the sections on root-water uptake and phenology. We feel that a greater discussion that specifically includes the 'Pulse-Reserve paradigm' is somewhat tangential to the key arguments we are making regarding the 3 processes we have centred the paper around.

**Comment 7:** Section 3.1: Possibly also mention potential for additional measurements to inform root water uptake dynamics (maybe around L590): -experimental use of isotopes to trace root water uptake dynamics (see work of Todd Dawson's lab for example). - standard field-measured sapflow and leaf gas exchange are surprisingly not mentioned but can be particularly useful when coupled with detailed soil moisture profile measurements, where changes over time directly indicate the effects of water uptake. -weighing lysimeter studies, while very intensive, have also been used to detect whole plant uptake. -groundwater wells would also be enormously helpful and are so often missed in ecological and even hydrological studies in savannas (and other ecosystems), yet are critical for characterizing the availability and dynamics of deep water sources. - groundwater maps, where available, are low hanging fruit for incorporation into spatial applications of TBMs. -another key thing that is missing is detailed mapping of C3 and C4 vegetation types (grasses/herbaceous), and their separate phenologies. -remotely sensed surface temperature is another valuable constraint on ecosystem water status (I think Damian Bonal was working on this and published on it).

**Author response:** We have added many of the suggested edits. See the response to comment 1 of reviewer #2

**Comment 8:** Conclusions go uncomfortably beyond what is supported in this paper and stray from the paper's clear focus on how to improve TBM performance for savannas. For example:

> "*Projected higher temperatures and rainfall variability, potentially promoting more frequent fires, could favour C4 grasses in mesic savanna, while drier conditions are expected to increase tree mortality in semi-arid savanna. Conversely, increases to atmospheric CO2 are expected to favour C3 trees, reflecting woody encroachment that is already observed in many savannas globally (Donohue et al., 2009). Climate change therefore has the potential to alter the carbon balance, which may have major feedbacks on global climate and biogeochemical cycling.*"

**Author response:** We have now deleted this text

**Comment 9:** Again, it is recommended that the authors expand the scope of highlights to also emphasize ecosystem structural and compositional dynamics that are of central importance to TBM processes: particularly differential resource acquisition (primarily water) and competitive interactions. E.g. around L694... model and data efforts should also target those attributes of savannas. Perhaps the authors roll all of that into "phenology" but I'd argue that this is a mistake, where phenology is only one component of vegetation dynamics. The underlying competitive interactions, mortality and growth dynamics, and how these shift in response to a suite of climate, atmospheric compositional, soil fertility, land use and other global change factors could receive more attention in this review.

**Author response:** We refer back to our responses to comments 2 and 3, and maintain that what the reviewer is arguing for here is outside the scope of the paper's aim. Differential resource acquisition, savanna structure and composition could be seen as emergent properties of the 3 dynamic processes this paper is focussed on improving. Once these first-order processes are better represented, then a subsequent investigation could be conducted into savanna structure and composition. Representation of competitive interactions would require an entirely different level of model complexity, e.g. individual-based models, which is not the subject of this paper.

**Some Details:**

Why is root-water hyphenated? Do you mean ground-water or soil-water? Probably just drop the hyphen throughout.

**Author response:** We are referring to soil-water that is within the rhizosphere. The decision to hyphenate root-water is purely stylistic, following the style adopted in other topically related papers in the literature. We are happy to defer to the editor's judgement on this.

**Line 69+:** not just "environmental conditions" but also biophysical and ecological conditions... that is, the ecosystem properties are themselves changing and this must be represented.

**Author response:** Done

**Line 96:** "confronting task" reword, unclear

**Author response:** Done

**L100:** "underperformed for savanna ecosystems" is too vague... what, specifically, lacks accuracy? "under" relative to what, other PFT or biome types, compared to data?

**Author response:** The term *underperformed* relates to statistical performance and refers to model error (difference between observation and prediction). This term is commonly used in the ecosystem modelling literature and references have been cited that go into this level of detail.

**L105:** "physical [and biological]"... most of these are not physical parameters.

**Author response:** We use the term physical in the context that they are not purely empirical or statistical; i.e. they are not arbitrary coefficients from a regression or polynomial equation.

**References:**

[revised manuscript text omitted]

---

## Author Response (AR2)

**Author responses to reviewer and editor comments for manuscript submission BG-2016-190: Reviewers 1-3, Assoc. Editor**

Below we outline our responses to the comments from reviewers for our paper entitled: "**Challenges and opportunities in modelling savanna ecosystems.**"

We thank all reviewers for their valuable feedback and insights. Both reviewers #1 and #2 provided only minor corrections, which we are largely in agreement with and have consequently adjusted the text of the paper to reflect this. Reviewer #3 has provided valuable and extensive commentary and insight on savanna ecology that is appreciated and along with recommendations from the editor we have substantially added a strong theoretical background to our review. We feel now that the paper is comprehensive and will appeal to a wider audience thanks to the reviewer's suggestions. At the same time we use the introductory sections (1 and 2) to cover the nature of savanna and many of the features that differentiate them globally. This paper builds on previous work (this special issue: Whitley et al. (2016)) that highlighted a set of processes related to savanna dynamics that are currently deficient in TBMs. In this paper, we continue to focus our discussion on how these processes are currently misrepresented (or absent) in TBMs and offer recommendations on how they could be developed to improve the predictive capability of such models in simulating the turbulent fluxes of savannas.

We have primarily focused discussion on three dynamic processes: i) phenology, ii) root water uptake and iii) disturbance (particularly fire), which are the first-order controls on savanna water and carbon exchange and should therefore be critical areas of future model development. Reviewer #3 raises issues regarding the lack of discussion relating to ecological processes such as tree-grass demographics, canopy structure, pulse responses to rainfall, etc. However, they could be seen as emergent behaviour resulting from the dynamical processes highlighted in this paper, and an expanded discussion of these issues could distract from what we consider the primary deficiencies that TBMs currently face in simulating savanna ecosystems. Furthermore, not all TBMs have the capability (or goal) of simulating complex vegetation dynamics. This is not to say these ecological properties of savannas are not important, on the contrary we believe that TBMs need to be able to replicate these effects, but this would be a consequent step after the first order processes we have highlighted have been improved. Nevertheless, we have included most of reviewer #3's suggestions. We also wish to stress that reviewer #3 has raised important issues ubiquitous within savanna ecology that could serve as a basis for future work and would serve as natural progression to what we have presented here.

Reviewer comments are numbered below, where we have answered each to the best of our

ability and made the appropriate changes where necessary in our manuscript. Once again, we would like to thank the reviewers and the editor for taking the time to examine this work and provide valuable feedback.

**Anonymous Reviewer #1**

**L51:** Remove "current-generation" as one might read this that previous generations were immune this challenge.

> **Author response:** Done.

**L60:** Remove ",namely"

> **Author response:** Done.

**L67:** Try "the effects"

> **Author response:** Done.

**L78:** Something is off here as "and provide important in providing ecosystem services,..." makes no sense.

> **Author response:** Has been corrected to: "... and **are** important in providing ecosystem services..."

**L84:** Try "creates"

> **Author response:** "and create demographic ..." has been changed to "that create demographic..."

**L88:** The antecedent of "it" is unclear, use "fire" again here.

> **Author response:** Done

**L96:** I think you want "confounding" here?

> **Author response:** Done

**L102:** Try "the current generation of TBMs has..."

    **Author response:** Done

**L126:** I think you "proceed" given that you use present tense throughout here.

    **Author response:** Done

**L184:** Remove the first "region"

    **Author response:** Done

**L189:** Replace "For the" with "As an"

    **Author response:** Done

**L190:** Try "to emerge"

    **Author response:** Done

**L207:** Replace "...occupy the top ranks among terrestrial biomes, together contributing c. 30%" with "...contribute c. 30%"

    **Author response:** Done

**L243:** Try "are"

    **Author response:** Done

**L246:** "until"? Until what?

    **Author response:** Sentence is complete now: "...until *later in this* paper."

**L268:** Try "are"

    **Author response:** Done

**L310:** Try "partition" and "LAI."

    **Author response:** Done

**L335:** Remove comma after "advances". Also, I must state that the paper needs a good final proofreading. I have pointed out several (albeit minor) issues but have certainly not caught all

the comma issues, and sundry other language faux pas.

**Author response:** Done, and furthermore we have gone through the text again to identify other typos and grammatical issues.

**L567:** What is NATT? Maybe define in L560 above.

**Author response:** The acronym has now been correctly added after the definition given at the beginning of the paragraph.

**L572:** Regarding your "as they cannot capture..." comment. I would dispute this especially as you invoke the space for time argument above. FLUXNET can quite do the same thing.

**Author response:** The reviewer is quite correct, and we have qualified this statement to say: "as they cannot *completely* capture..."

**L575:** Citations are off.

**Author response:** Fixed

**L591:** I appreciate that the authors can't solve all these data limitations. But the "such data may be critical" comment is an interesting one, especially in the context of rather dear excavation studies. I'd like more detail. How many such excavation studies with what sampling design frame do you envision? That is, how do we move forward as a community to actually get the right data?

**Author response:** Root excavation studies as mentioned was only given here in a general sense as an example, however we see the reviewers point that such field campaigns are complex and expensive (in cost, time and labour), such that this warrants a further expansion of detail. We have therefore added the following lines to qualify our statement not just for root excavations, but also for all ecological trait information:

*"We recommend that future EC studies, particularly along transects as mentioned above, should include intensive field campaigns that are targeted towards a more complete characterisation of the site. This would include key flux measurements (e.g. sapflow, stomatal conductance, leaf water potentials, deep soil water measurements, root excavations and the collection of plant trait data (e.g. leaf mass per area, capacitance, Rubisco activity, etc.) within the footprint of an EC tower. Collaborative research networks, such as TERN (Terrestrial Ecosystem Research Network), NEON (National Ecological Observatory Network) and SAEON (South African Environmental*

*Observation Network) that have the resources and infrastructure to conduct such campaigns will be instrumental to meet these demands for more observational data."*

**L603:** In this section I would encourage the authors to cite some other developments here, e.g., ILAMB, that certainly hold promise to improve benchmarking. PALS is well and good but there is more afoot.

> **Author response:** We agree with the reviewer and have incorporated mention of ILAMB and other model intercomparison projects (e.g. PILPS, C4MIP) into this part of the discussion.

**L692:** Might NEON be a good idea? I must say I've noted a rather Australian-centric view of the literature. That is not bad, particularly in an OzFLux special issue, but again there are other things afoot and this is a review paper. And savannas do not exist solely in Australia.

> **Author response:** We have now included mention of NEON as well as SAEON at the end of Section 3.1 (see response to comment on L591)

**L703:** I am confused on the juxtaposition of long-term EC sites and fire return. A fire typically has adverse consequences for a FLUXNET installation. Are you advocating pre- and post-fire EC measurements?

> **Author response:** Fire is a a frequent occurrence in savanna and has a major impact on fluxes and we propose that savanna FLUXNET installations quantify the effects of fire as per Beringer et al. (2007) and as was mentioned in Section 2.3. In this regard we are advocating pre- and post-fire measurements, as this would allow TBMs (those that include the simulation of fire) to be tested on whether they have the capability to simulate the nonlinear response of the canopy (due to scorching and reduced surface albedo) during the post-fire recovery period.

**Anonymous Reviewer #2**

My minor comments are:

**Comment 1:** I found the description of additional dataset (ancillary and remote sensing data) which can be used to test the ecological model a little bit unclear. However, this part can be easily

improved by the authors by adding more details on the **variables** which can be extrapolated from these datasets at the end of the session on "Datasets to inform model development" (P18, L583).

**Author response:** We have now made reference to common model parameters that would benefit from the collection of specific environmental information. This text is quoted below as:

*"Digital soil atlases also provide an excellent resource in parameterising simulated soil profiles (e.g. Isbell, 2002; Sanchez et al., 2009). However, the spatial resolution of these data products can be coarser than the operating resolution of many TBMs, such that site-level measurements should be used when possible. Excavation studies that quantify savanna tree root-systems (Chen et al., 2004) and soil-moisture probes installed at greater depths (> 2 m) are informative about the evolution of the soil-root zone over time (e.g. surface root density, root depth), and such data may be critical to understanding whether current root-water extraction schemes in TBMs are capable of simulating the dry season response of savanna tree species (Whitley et al., 2015). Finally, localised observations of plant traits such leaf-mass per area, leaf capacitance, tree height, etc. are needed to inform a better parameterisation of savanna specific PFTs (Cernusak et al., 2011). For example, specific leaf-level information such as Rubisco activity ($V_{cmax}$) and RuPB regeneration ($J_{max}$) for both $C_3$ and $C_4$ plants are critically needed to inform the Farquhar leaf photosynthesis models (Farquhar et al., 1980), while information on $g_s$ and leaf water potential ($\Psi_{leaf}$) are important in parameterising the many stomatal conductance models used in TBMs (Ball et al., 1987; Medlyn et al., 2011; Williams et al., 1996). Leaf capacitance and water potential data are also critically important in characterising model sensitivity to drought (Williams et al., 2001), but this information is severely lacking for savannas."*

**Comment 2:** A short discussion on the scale mismatch between these datasets (including EC) and the model grid should be added in the "Model evaluation and benchmarking".

**Author response:** We agree that the intention of almost all TBMs is to be run at the global scale does not match the scale at which validation occurs. Model evaluation of TBMs occurs at the ecosystem scale (a moderate resolution of ~1 km), commensurate with resolution of flux tower data and remotely sensed data products (e.g. Best et al., 2015; Blyth et al., 2010). Consequently, there is little scale mismatch between what is used to run (inputs) and validate (outputs) the models.

**Specific comments**

**P5, L139:** Please substitute "ecosystem types" with "ecoclimate regions"

> **Author response:** Done

**P7, L183:** Please define the two acronyms: LSM and DGVM

> **Author response:** Done

**P9, L252:** This should be section "3" and not section "2". Please check and re-number, where needed, all sections

> **Author response:** Done. All section numbering has been updated accordingly.

**P9, L252:** Please eliminate ":" and the end of the title

> **Author response:** Done

**P17, L524:** Please eliminate "ground-based"

> **Author response:** Done

**P17, L531-534:** This sentence is not very clear for people that don't know very well how models and the eddy covariance system work. Please rephrase

> **Author response:** We have changed the sentence in accordance with this request:
>
> "*Turbulent fluxes measured by EC systems that include net ecosystem exchange and latent and sensible heat are common model outputs, such that this information is commonly used to validate TBMs. Local meteorological forcing (e.g. short-wave irradiance; SW, air temperature, rainfall, etc.) that is concurrently measured with the turbulent fluxes by other instruments (rainfall and temperature gauges, radiation sensors, etc.) are common model inputs and are used to drive TBMs.*"

**P17, L534:** I would like to use "ecosystem scale" instead of "spatial scale" to better give the idea of the spatial representativeness of EC data which are limited to the footprint area

> **Author response:** Done

**P17, L534:** Please refer to Aubinet et al., 2012

Aubinet, M., Vesala, T., and Papale, D.: Eddy Covariance – A Practical Guide to Measurement and Data Analysis, Springer, ISBN: 978-94-007-2351-1, 2012

> **Author response:** Done

**P17, L535:** Please add reference: Balzarolo et al., 2014; doi:10.5194/bg-11-2661-2014

> **Author response:** Done

**P18, L567:** Please explain NATT⟦SEP⟧P18, L583: Please revise this sentence, which is not very clear

> **Author response:** Definition has now been inserted, and sentence now reads as:
>
> *"A recent model intercomparison study by Whitley et al. (2015) used turbulent flux observations sampled along the NATT to evaluate a set of six TBMs, and documented only poor to moderate performance for those savanna sites."*

**P18, L573:** Are you referring to long-term temporal predictions?

> **Author response:** No, what we are alluding to in this sentence is that eddy-covariance systems may not have been running for long enough (i.e. the sampling period of the time-series) at a site to pick up any long-term structural changes (e.g. such as those caused by cyclones, or large fires). If a model is attempting to simulate a demographic shift in vegetation (i.e. the tree/grass ratio) then longer term datasets such as those from satellite would be more useful in validating this prediction.

**P21, L680:** Please refer to Fluxnet (Baldocchi et al., 2001)

Baldocchi, D. D., Falge, E., Gu, L., Olson, R., Hollinger, D., Running, S., Anthoni, P., Bernhofer, C., Davis, K., Fuentes, J., Goldstein, A., Katul, G., Law, B., Lee, X., Malhi, Y., Meyers, T., Munger, J. W., Oechel, W., Pilegaard, K., Schmid, H. P., Valentini, R., Verma, S., Vesala, T., Wilson, K., and Wofsy, S.: FLUXNET: a new tool to study the temporal and spatial variability of ecosystem-scale carbon dioxide, water vapor and energy flux densities, B. Am. Meteorol. Soc., 82, 2415–2435, 2001.

> **Author response:** Done

**P22, L693:** Please also refer to NEON (National Ecological Observatory Network)

**Author response:** Done

**Figure 1:** please add (a) and (b) in the figure and check the appropriate units to the y-axis (mean annual rainfall is correct in mm/month?)

**Author response:** Done, the units are correct but we have rescaled the values to be mm/year.

**Figure 2:** it is not very clear the use of "2012" and "2011" in the legend. Please change or explain in the caption.

**Author response:** Done.

**Anonymous Reviewer #3**

The reviewer points that we have omitted "some important theory and themes in savanna ecology, and misrepresents some of the broad geographic context of global savannas". Given the previous aims we consciously did not attempt an 'ecological' review that could be a distraction from the main paper. However, given the opportunity to review the paper we have now provided a clear overview of the global savanna ecology and this has been included in the text. We have added considerable material and therefore do not include the specific changes below but rather show them changes and additions as track changes in the supplied manuscript.

*Note: we have removed the listing of references for brevity, but can be referred to in the reviewers original commentary.*

**Comment 0:** Regarding Root Water Uptake: This is a worthwhile focus for improvements of TBMs but the authors have a rather specific read on the relevant literature. A few key citations I'd recommend are below, offering expanded perspectives on how to proceed with improving root water uptake in TBMs. Key considerations go beyond just prescribing rooting depth but also: dynamic uptake in response to soil water availability in the vertical profile, adaptive adjustments of the root distributions in response to water availability over seasonal and multi-year timescales, hydraulic redistribution along pressure gradients and via roots, soil water limitation function limiting productivity and evapotranspiration and associated water demand and water potential along the root to leaf and atmosphere continuum. I, too, caution against weighting root water up- take by fine root distribution because many plants are able to sustain water uptake and transpiration from deep taproots that access the saturated zone or deep

unsaturated zone water sources even when fine roots (all concentrated near the surface) are in very dry soil layers. Furthermore, plant capacitance (storage) is important for accurately representing plant water potential, especially when water supply is limited or when root-to-leaf transport resistances inhibit water delivery to the site of transpiration at the leaf.

**Author response:** We agree that other belowground processes the reviewer has mentioned are critical to simulating integrated representation of the soil-vegetation-atmospheric continuum (SVAC). Section 2.2 on root-water access and uptake does cover most of the processes the reviewer has mentioned, particularly dynamic uptake. We have also added a paragraph to capture the other dynamic ecological processes that could be important along with additional material as found in the track changes manuscript. It should be said that the aim of this section is to argue for a better representation of root water access in and around the rhizosphere, as this is describes the supply end of the SVAC. Consequently it acts as a first order control for savanna water and carbon exchange in the dry season, which has not been modelled well in past studies. Hence we have specifically focused our discussion on this limitation, and that the modelling community could easily address this. Regarding plant capacitance, we agree that this is a critical parameter towards vegetation drought responses and have amended the second sentence of the first paragraph of Section 2.2 to reflect this.

> *"For seasonally dry climates (a fundamental characteristic of savanna ecoclimatic regions), productivity is primarily limited by dry season water-availability (Kanniah et al., 2010), which is largely determined by plant regulation of water transport (through leaf stomatal conductance and stem capacitance) and the root zone water storage capacity and access (distribution of fine root biomass)."*

**Comment 1:** A host of other processes of importance and interest in savannas are missed. For example, pulse response processes, stand-scale vegetation composition, plant-scale competitive interactions, stand-scale vegetation structure, landscape patterning of vegetation, nutrient cycling and interactions with herbivores, and more are all given little if any attention. Arguably many of those processes are important for representing savanna-atmosphere interactions, and for assessing savanna responses to global change factors. Given that this paper is intended to be a review of key processes that need to be considered to accurately model savanna ecological responses to global change factors, I would encourage additional discussion of these missed processes and their implications and importance for the stated aims.

**Author response:** We agree that these are important processes in savanna ecology, however

and we have taken the opportunity to mention these throughout the manuscript. We also make a comment that the *stand-scale vegetation composition, plant-scale competitive interactions, stand-scale vegetation structure, etc.* can be seen as emergent properties of the three dynamic processes that our paper focuses on. Given that the discussion is focussed on the first order controls on savanna water and carbon exchange and how these are likely misrepresented (or absent) from modern TBMs we concentrate on how TBMs can better simulate savanna intra- and interspecies competition, canopy structure, etc. the 3 first-order dynamic processes as a priority..

**Comment 2:** Savanna ecologists would be underwhelmed by the three dynamic processes that are highlighted: phenology, root water uptake, and fire, given that these have long been the focus of their work going back many decades (e.g. Walter 1973). For example, the seminal work by Brian Walker (1981) is surprisingly absent from the present review even though this was foundational work identifying the importance of root zone separation and differential uptake zones for grass/herbaceous and woody PFTs in the savanna matrix. This was nicely tested in the Scholes and Walker (1993) book which is also missed. It is surprising that competitive interactions and differential resource access are not noted here, nor differential response of PFTs and species to single and multifactor drivers of CO2, drought, warming, and increasing VPD. While I agree that the three features highlighted in the present paper are essential and yet poorly represented (if at all) in TBMs, such models will still not be up to the task of predicting responses to global changes without representation of a host of other factors.

**Author response:** We are fully aware of the decades of ecological research on the three dynamic processes that are highlighted in our review section; phenology, root water uptake, and fire, but these have been poorly captured in TBMs to date, and this is the focus of this paper. Our goal is not to provide a review of determinants of savanna structure and function but to target the failing of TBMs. However, we have also added material that points to the many other dynamic processes that could be influential.

**Comment 3:** Grazing and browsing are of central importance in many of the world's savannas, strongly influencing vegetation cover, loss of productivity and biomass, species com- position, and affecting site fertility but this driver is hardly mentioned, receiving just one or two sentences. A bit more on this subject would seem warranted for such a review.

**Author response:** We certainly agree that grazing and browsing are one of many important factors that modulate the savanna structure and state. However, animals are not a modelled process in TBMs and this paper is focussed on simulated processes that control savanna water and carbon exchange. Nevertheless, we have added material in the background to global savannas.

**Comment 4:** Corresponding to the above, a nod to the alternate stable states literature is missing here, including Walker '81, Noy-Meir 1975, and others mentioned below.

**Author response:** We have now added a section describing these meta-stable systems into the main text.

**Comment 5:** Discussion of the global context and diversity of savanna attributes and strategies is lacking and in some ways misleading. Section 2.1, particularly Line 169+: The language here misrepresents the growth and longevity strategies of woody plants in Africa. Many of the woody species in at least southern Africa do indeed have deep roots but groundwater is deep (probably deeper than in much of Australia) so there is less potential to rely on near surface (<10 m) water sources. The Archibald and Scholes '97 paper does not mention roots once, and says nothing about strategies of water access. The Higgins '11 paper also offers little on root water uptake. Both of those papers do indeed discuss and quantitatively document phenological dynamics, but neither indicates that the full woody component of southern African savannas is deciduous (indeed Acacia sp. often retain leaves consistently through the dry season). However both represent only southern African ecosystems at best (really, Kruger Park). Yet this statement is as grandiose as generalizing from these studies to all African and South American savannas! That's stretching it a bit, no? A much broader literature must be invoked if the authors truly want to discuss geographic patterns of root water uptake, and diversity in savanna traits and properties. Furthermore, this must consider not just phenology but also water availability in the unsaturated and saturated zones, and not confuse mesic and arid savanna types. The present interpretation seems to conflate shallow groundwater availability or its absence with a difference in plant strategy. However, woody species of savannas around the world "favour a long-term strategy of conservative growth that is insured against an unpredictable climate", not just those in Australia. To include more on the global biogeography of savannas relevant to a modeling context I'd recommend some additional reading (and citation of) works in: Hill, Michael J. and Hanan, Niall P. eds (2011). Ecosystem Function in

Savannas: Measurement and Modeling at Landscape to Global Scales. (CRC Press, Boca Raton, Florida) 559 pp.

**Author response: We have now added material covering the** global context and diversity of savanna attributes and strategies found in the track changes document. We have also included citations from Bowman and Prior (2005), Lehmann et al. (2011, 2014) and Stevens et al. (2016) that support our argument that woody species among continents can be clustered as we have stated in the text. The Archibald and Scholes (1997) is a mistake and should be Scholes and Archer (1997), now corrected, while Higgins et al. (2011) has been removed given that it does not directly support our claims. We do agree that not all African woody species are shallow-rooted and have qualified the statement to read as:

*"Canopies of the African and South American savanna regions are predominantly characterised by deciduous woody species that are in most cases (although not always) shallow-rooted and follow a short-term growth strategy that maximises productivity while environmental conditions are favourable"*

We do not believe that expanding the discussion on global savanna attributes to cover every degree of separation of plant traits between and within continents benefits our argument on the 3 dynamics processes that we believe currently hamper TBM performance in savanna ecosystems. The primary point of this section is to highlight and provide context that these differences suggest that savannas cannot be lumped into some generalised group or plant functional type (PFT), but show clear distinctions. Rather, we argue that region specific PFTs will likely be required for good model performance in this ecosystem. However, this is not to say that the assignment of correct trait information is the answer to improved model performance in savannas, rather a better representation of phenology, root-water uptake and disturbance (fire) in TBMs is required to fill this gap.

**Comment 6:** The Pulse-Reserve paradigm in dryland ecology is noticeably absent from this review despite the well-known importance of rainfall pulses in organizing complex ecological and biophysical dynamics in water-limited environments. Many plant and ecosystem phenological dynamics are organized around rainfall pulses, including leaf-out and senescence, up- and down-regulation of productivity, respiration and decomposition bursts, reproduction and establishment events, and so forth. "Pulse" is not mentioned once in the current review.

**Author response:** We agree that the ecosystem response to rainfall pulses in arid climates is an

important behaviour for TBMs to capture. We have now provided some references and links to the importance of pulse dynamics in the track changes version of the document However, TBMs should have the capability to capture this response, and if not, it is likely not a missing mechanism per se, rather a lack of sensitivity. We have discussed issues related to this response in the sections on root-water uptake and phenology.

**Comment 7:** Section 3.1: Possibly also mention potential for additional measurements to inform root water uptake dynamics (maybe around L590): -experimental use of isotopes to trace root water uptake dynamics (see work of Todd Dawson's lab for example). - standard field-measured sapflow and leaf gas exchange are surprisingly not mentioned but can be particularly useful when coupled with detailed soil moisture profile measurements, where changes over time directly indicate the effects of water uptake. -weighing lysimeter studies, while very intensive, have also been used to detect whole plant uptake. -groundwater wells would also be enormously helpful and are so often missed in ecological and even hydrological studies in savannas (and other ecosystems), yet are critical for characterizing the availability and dynamics of deep water sources. - groundwater maps, where available, are low hanging fruit for incorporation into spatial applications of TBMs. -another key thing that is missing is detailed mapping of C3 and C4 vegetation types (grasses/herbaceous), and their separate phenologies. -remotely sensed surface temperature is another valuable constraint on ecosystem water status (I think Damian Bonal was working on this and published on it).

**Author response:** We have added many of the suggested edits. See the response to comment 1 of reviewer #2. In addition we have added the following into the section on "*Datasets to inform model development";* Other useful approaches for elucidating how and where plants gain their water, include sap flow measurements (Zeppel et al., 2008), gas chambers (Hamel et al., 2015) and soil-plant-water experiments (Midwood et al., 1998). In additional, hydrogen and oxygen stable isotope ratios of water within plants provide new information on water sources, interactions between plant species and water use patterns under various conditions (see review by Yang et al. (2010)).

**Comment 8:** Conclusions go uncomfortably beyond what is supported in this paper and stray from the paper's clear focus on how to improve TBM performance for savannas. For example:

"*Projected higher temperatures and rainfall variability, potentially promoting more frequent fires, could favour C4 grasses in mesic savanna, while drier conditions are expected to increase tree mortality in semi-arid savanna. Conversely, increases to atmospheric CO2 are*

*expected to favour C3 trees, reflecting woody encroachment that is already observed in many*
*savannas globally (Donohue et al., 2009). Climate change therefore has the potential to alter the*
*carbon balance, which may have major feedbacks on global climate and biogeochemical cycling."*

**Author response:** We have now deleted this text

**Comment 9:** Again, it is recommended that the authors expand the scope of highlights to also emphasize ecosystem structural and compositional dynamics that are of central importance to TBM processes: particularly differential resource acquisition (primarily water) and competitive interactions. E.g. around L694... model and data efforts should also target those attributes of savannas. Perhaps the authors roll all of that into "phenology" but I'd argue that this is a mistake, where phenology is only one component of vegetation dynamics. The underlying competitive interactions, mortality and growth dynamics, and how these shift in response to a suite of climate, atmospheric compositional, soil fertility, land use and other global change factors could receive more attention in this review.

**Author response:** We refer back to our responses to comments 2 and 3, and maintain that what the reviewer is arguing for here is outside the scope of the paper's aim. Differential resource acquisition, savanna structure and composition could be seen as emergent properties of the 3 dynamic processes this paper is focussed on improving. Once these first-order processes are better represented, then a subsequent investigation could be conducted into savanna structure and composition. Representation of competitive interactions would require an entirely different level of model complexity, e.g. individual-based models, which is not the subject of this paper. Nevertheless, we do now acknowledge these in terms of future model development

"There is still great uncertainty in predicting the future of savanna biomes (Scheiter et al., 2015; Scheiter and Higgins, 2009) and improving how savanna ecosystems are represented by TBMs will likely encompass the consideration of additional processes that have not been mentioned here. This will no doubt include improved understanding of ecological theory that will lead to improvements in modelling ecosystem demographics and tree-grass interaction that will improve DGVMs."

**Some Details:**

Why is root-water hyphenated? Do you mean ground-water or soil-water? Probably just drop

the hyphen throughout.

**Author response:** We are referring to soil-water that is within the rhizosphere. The decision to hyphenate root-water is purely stylistic, following the style adopted in other topically related papers in the literature. We are happy to defer to the editor's judgement on this.

**Line 69+:** not just "environmental conditions" but also biophysical and ecological conditions... that is, the ecosystem properties are themselves changing and this must be represented.

**Author response:** Done

**Line 96:** "confronting task" reword, unclear

**Author response:** Done

**L100:** "underperformed for savanna ecosystems" is too vague... what, specifically, lacks accuracy? "under" relative to what, other PFT or biome types, compared to data?

**Author response:** The term *underperformed* relates to statistical performance and refers to model error (difference between observation and prediction). This term is commonly used in the ecosystem modelling literature and references have been cited that go into this level of detail.

**L105:** "physical [and biological]"... most of these are not physical parameters.

**Author response:** We use the term physical in the context that they are not purely empirical or statistical; i.e. they are not arbitrary coefficients from a regression or polynomial equation.

**References:**

[revised manuscript text omitted]

We have now implemented changes as required by R3 (described above), plus implemented directions as provided by the Associate Editor of the OzFlux Special Issue ('Reconsider after major revisions, 13 Mar 2017').

The title of the paper has been modified from '*Challenges and opportunities in modelling savanna ecosystems*' to "*Challenges and opportunities in land surface modelling of savanna ecosystems*".

Text in the Introduction provides a stronger focus on the performance of modelling land-atmospheric fluxes in savannas. This revised version is labelled Ver 3.1.

These modifications have been implemented to focus the review on challenges associated with modelling fluxes from savanna ecosystems, as opposed to pitching the review at addressing 'whole of ecosystem' models and associated processes. The definition of the suite of models we are referring to, Terrestrial Biosphere Models (TBMs), has been revised and introduced far earlier in this revised ms (Line 61).

We have provided a more comprehensive description of ecological theory and conceptual models of savanna structure and function, as required by R3, this has been further revised in this version, over and above what we presented in Ver 3.0. The 'Savanna biome' section, Section 2 (Line 120 – L274), now has 4 sub-sections entitled: *2.1 Characteristics and global extent; 2.2 Conceptual models of tree and grass co-existence; 2.3 Determinants of savanna structure and 2.4 Potential impacts of climate change*.

This section better informs our identified areas of model failure; inadequate descriptions of rooting depth, and impacts from herbivory and disturbance. We conclude with a statement that future model development is required to better capture these processes that are particular to savanna ecosystems. This will result in more accurate modelling of primary productivity, water use and energy exchange for this extensive biome.

The corresponding author has been changed on the ms to Jason Beringer (UWA). Whitley remains as the lead author. Changes have been tracked. No figures required further modification in our view.